# The ketone body β-hydroxybutyrate ameliorates neurodevelopmental deficits in the GABAergic system of *daf-18/PTEN Caenorhabditis elegans* mutants

**Sebastián Giunti[1,2], María Gabriela Blanco[1,2], María José De Rosa[1,2]\*, Diego Rayes[1,2]\***

[1]Instituto de Investigaciones Bioquímicas de Bahía Blanca (INIBIBB) (UNS-CONICET), Universidad Nacional del Sur-Consejo Nacional de Investigaciones Científicas y Técnicas, Bahia Blanca, Argentina; [2]Departamento de Biología, Bioquímica y Farmacia, Universidad Nacional Del Sur (UNS), Bahia Blanca, Argentina

**\*For correspondence:**
mjderosa@criba.edu.ar (MJDR);
drayes@criba.edu.ar (DR)

**Competing interest:** The authors declare that no competing interests exist.

**Abstract** A finely tuned balance between excitation and inhibition (E/I) is essential for proper brain function. Disruptions in the GABAergic system, which alter this equilibrium, are a common feature in various types of neurological disorders, including autism spectrum disorders (ASDs). Mutations in *Phosphatase and Tensin Homolog (PTEN)*, the main negative regulator of the phosphatidylinositol 3-phosphate kinase/Akt pathway, are strongly associated with ASD. However, it is unclear whether *PTEN* deficiencies can differentially affect inhibitory and excitatory signaling. Using the *Caenorhabditis elegans* neuromuscular system, where both excitatory (cholinergic) and inhibitory (GABAergic) inputs regulate muscle activity, we found that *daf-18/PTEN* mutations impact GABAergic (but not cholinergic) neurodevelopment and function. This selective impact results in a deficiency in inhibitory signaling. The defects observed in the GABAergic system in *daf-18/PTEN* mutants are due to reduced activity of DAF-16/FOXO during development. Ketogenic diets (KGDs) have proven effective for disorders associated with E/I imbalances. However, the mechanisms underlying their action remain largely elusive. We found that a diet enriched with the ketone body β-hydroxybutyrate during early development induces DAF-16/FOXO activity, therefore improving GABAergic neurodevelopment and function in *daf-18/PTEN* mutants. Our study provides valuable insights into the link between *PTEN* mutations and neurodevelopmental defects and delves into the mechanisms underlying the potential therapeutic effects of KGDs.

## eLife assessment

This is a conceptually appealing study in which the authors identify genes whose function is **important** for the development of inhibitory (GABA) neurons, and then demonstrate that a diet rich in ketone body β-hydroxybutyrate partially suppresses specific mutant phenotypes. The authors provide **compelling** evidence that features methods, data and analyses more rigorous than the current state-of-the-art. Conceptually, this is evidence of a rescue of a developmental defect with dietary metabolic intervention, linking, in an elegant way, the underpinning genetic mechanisms with novel metabolic pathways that could be used to circumvent the defects.

**eLife digest** To work optimally, the brain needs to delicately balance excitation and inhibition – that is, it must precisely control exactly when and how excitatory neurons (which activate the system) or inhibitory ones (which counteract these activating signals) are switched on. Neurological disorders can arise when this equilibrium is disrupted, for example when defects are present in an inhibitory signalling system that relies on a molecule known as GABA. More recently, a gene known as *PTEN* has also emerged as playing an important role during the development of the nervous system, yet exactly why this is the case has remained unclear.

To explore this question, Giunti et al. focused on the neuromuscular system of the roundworm *Caenorhabditis elegans*, in which excitatory ('cholinergic') and inhibitory ('GABAergic') neurons control how muscles contract and relax. A range of biological approaches were used to assess the impact of PTEN deficiencies on this system. This revealed that mutations in this gene do not impact cholinergic activity; they did, however, lead to diminished GABAergic activity. Overall, this resulted in an increased ratio of excitatory to inhibitory activity in the system.

Further work showed that, in the mutated worms, the suppression of inhibitory neurons was due to a specific protein being inactive during early development. This transcription factor is the worm equivalent of the human FOXO protein, and it helps to turn genes on and off during development. Its inactivity is linked to noticeable changes in the shape and activity of GABAergic neurons.

In humans, medical ketogenic diets (which force the body to use fats rather than sugars as a source of energy) are known to improve conditions linked to imbalances in the excitatory and inhibitory systems. Giunti et al. therefore investigated whether a similar approach could mitigate some of the defects seen in PTEN mutants. Exposing these worms early in development to a type of molecule produced in ketogenic diets partly improved the state of their GABAergic neurons.

Taken together, this work suggests a potential molecular basis for the association between *PTEN* and the balance between excitatory and inhibitory activity. As *PTEN* mutations are often found in individuals diagnosed with autism spectrum disorders, further research is necessary to validate these findings in mammals and explore their clinical relevance.

## Introduction

Maintaining a delicate balance between excitatory and inhibitory (E/I) neurotransmission is critical for optimal brain function (*Tao et al., 2014*). Disruptions in this balance are commonly observed in neurodevelopmental disorders (*Bozzi et al., 2018*; *Coghlan et al., 2012*; *Oblak et al., 2009*). In particular, deficits in inhibitory (GABAergic) signaling have been reported in autism spectrum disorders (ASDs) and other related physiopathological conditions (*Oblak et al., 2009*; *Gogolla et al., 2009*).

*PTEN* is a classical tumor suppressor gene that antagonizes the highly conserved phosphatidylinositol 3-phosphate kinase (PI3K)/protein kinase B (PKB/Akt) pathway. Several reports using animal models have highlighted the importance of *PTEN* in neurodevelopment (*Rademacher and Eickholt, 2019*; *van Diepen and Eickholt, 2008*; *Kwon et al., 2006*; *Smith et al., 2016*; *Clipperton-Allen and Page, 2014*; *Chen et al., 2015*; *Clipperton-Allen et al., 2022*). Moreover, mutations in *PTEN* were frequently found in human patients presenting ASD (*Butler et al., 2005*). The molecular events underlying the neurodevelopmental deficits in *PTEN* mutants remain poorly understood.

The *Caenorhabditis elegans* neuromuscular system, where both excitatory (cholinergic) and inhibitory (GABAergic) motor neurons regulate muscle contraction and relaxation, provides an excellent platform for studying the function, balance, and coordination between excitatory and inhibitory signals (*Huang et al., 2019b*; *Vashlishan et al., 2008*; *Zhou and Bessereau, 2019*; *Safdie et al., 2016*; *Stawicki et al., 2011*; *Opperman et al., 2017*; *Giles et al., 2019*). This system has yielded valuable insights into fundamental synaptic transmission mechanisms (*Blazie and Jin, 2018*; *Calahorro and Izquierdo, 2018*). Over the last decade, numerous studies focused on this simple yet highly informative system have significantly contributed to our understanding of the functioning and dysregulation of human genes associated with neurodevelopmental disorders, epilepsy, and familial hemiplegic migraine (*Huang et al., 2019b*; *Opperman et al., 2017*; *Giles et al., 2019*; *Bessa et al., 2013*). Furthermore, the substantial conservation of the main components of the PI3K/Akt pathway in *C. elegans* (*Paradis and Ruvkun, 1998*; *Ogg and Ruvkun,*

*1998*) enhances the applicability of this model system for investigating the role of this pathway in neurodevelopment.

We here found that mutations in *daf-18* (the ortholog for *PTEN* in *C. elegans*) result in impairments in GABAergic inhibitory signaling due to decreased activity of the transcription factor DAF-16 (the ortholog for FOXO in *C. elegans*) during neurodevelopment. Interestingly, cholinergic excitatory motor neurons remain unaffected. This targeted impairment of inhibitory signals causes an imbalance between E/I neurotransmission in the animal's neuromuscular system.

In humans, ketogenic diets (KGDs), which force fatty acids beta-oxidation into ketone bodies, have been utilized for decades to treat pathologies associated with E/I imbalances, such as refractory epilepsies (*D'Andrea Meira et al., 2019*; *Neal et al., 2008*; *Lambrechts et al., 2017*). More recently, KGDs have also demonstrated effectiveness in alleviating autistic symptoms in humans (*Li et al., 2021*) and rodent models of ASD (*Castro et al., 2017*; *Verpeut et al., 2016*). The mechanisms underlying these beneficial effects remain largely unknown.

We demonstrated that exposing *daf-18/PTEN* mutants to a diet enriched with the ketone body β-hydroxybutyrate (βHB) early in development enhances DAF-16/FOXO activity, mitigates morphological and functional defects in GABAergic neurons, and improves behavioral phenotypes. This study not only provides a straightforward system for studying the role of the conserved PI3K/Akt/FOXO pathway in neurodevelopment but also contributes to our understanding of the mechanisms underlying the effects of ketone bodies in neurodevelopment.

## Results

### Mutants in *daf-18/PTEN* and *daf-16/FOXO* are hypersensitive to cholinergic drugs

Disturbances in *C. elegans* cholinergic or GABAergic activity can be detected by analyzing the sensitivity to the paralyzing effects of drugs that exacerbate cholinergic transmission (*Huang et al., 2019b*; *Vashlishan et al., 2008*). We analyzed the sensitivity of *daf-18/PTEN*-deficient animals to the acetylcholinesterase inhibitor aldicarb and the cholinergic agonist levamisole. Exposure to aldicarb leads to an increase in ACh levels at cholinergic motor synapses, resulting in massive activation of muscular cholinergic receptors and subsequent paralysis (*Mahoney et al., 2006*). Levamisole also induces paralysis by directly activating muscular cholinergic receptors (*Mahoney et al., 2006*; *Figure 1A*). We found that *daf-18/PTEN* mutants are hypersensitive to the paralyzing effects of both drugs (*Figure 1B, C*). Hypersensitivity to cholinergic drugs is typical of animals with an increased E/I ratio in the neuromuscular system, such as mutants in *unc-25* (the *C. elegans* ortholog for glutamic acid decarboxylase, an essential enzyme for synthesizing Gamma-Aminobutyric Acid [GABA]) (*Huang et al., 2019b*; *Vashlishan et al., 2008*). While *daf-18/PTEN* mutants become paralyzed earlier than wild-type animals, their hypersensitivity to cholinergic drugs is not as severe as that observed in animals completely deficient in GABA synthesis, such *unc-25* null mutants (*Figure 1B, C*) indicating a less pronounced imbalance between excitatory and inhibitory signals. Reduced activity of DAF-18/PTEN has been largely shown to exacerbate the PI3K pathway, precluding the activation of DAF-16/FOXO, the *C. elegans* ortholog of the FOXO transcription factors family (*Ogg and Ruvkun, 1998*; *Figure 1—figure supplement 1*). We analyzed aldicarb and levamisole sensitivity of mutants in this transcription factor. Similar to *daf-18/PTEN* mutants, we found that *daf-16/FOXO* null mutants are hypersensitive to the paralyzing effects of aldicarb and levamisole (*Figure 1C*). Furthermore, we did not observe significant differences in aldicarb and levamisole sensitivity between *daf-18;daf-16* double null mutants and the respective single mutants, suggesting that both genes affect neuromuscular signaling by acting in the same pathway (*Figure 1C*). In addition to *daf-16/FOXO* and *daf-18/PTEN*, we assessed the sensitivity to the paralyzing effects of aldicarb and levamisole in loss-of-function mutants of other components of the PI3K pathway, such as *age-1/PI3K*, *pdk-1*, *akt-1*, and *akt-2* (*Figure 1D*). Unlike mutations in *daf-18/PTEN*, in these mutants the PI3K pathway is downregulated (*Figure 1—figure supplement 1*). We did not observe significant differences compared to the wild-type. Due to the complete dauer arrest observed in double mutants of *akt-1* and *akt-2* (*Tao et al., 2014*; *Oh et al., 2005*; *Quevedo et al., 2007*), we were unable to explore the potential redundancy of these two genes. Interestingly, a gain-of-function mutant in *pdk-1*, *pdk-1(mg142)* (*Paradis and Ruvkun, 1998*) is hypersensitive to aldicarb and levamisole, similar to *daf-16/FOXO* and *daf-18/PTEN* mutants (*Figure 1D*). Given that increased

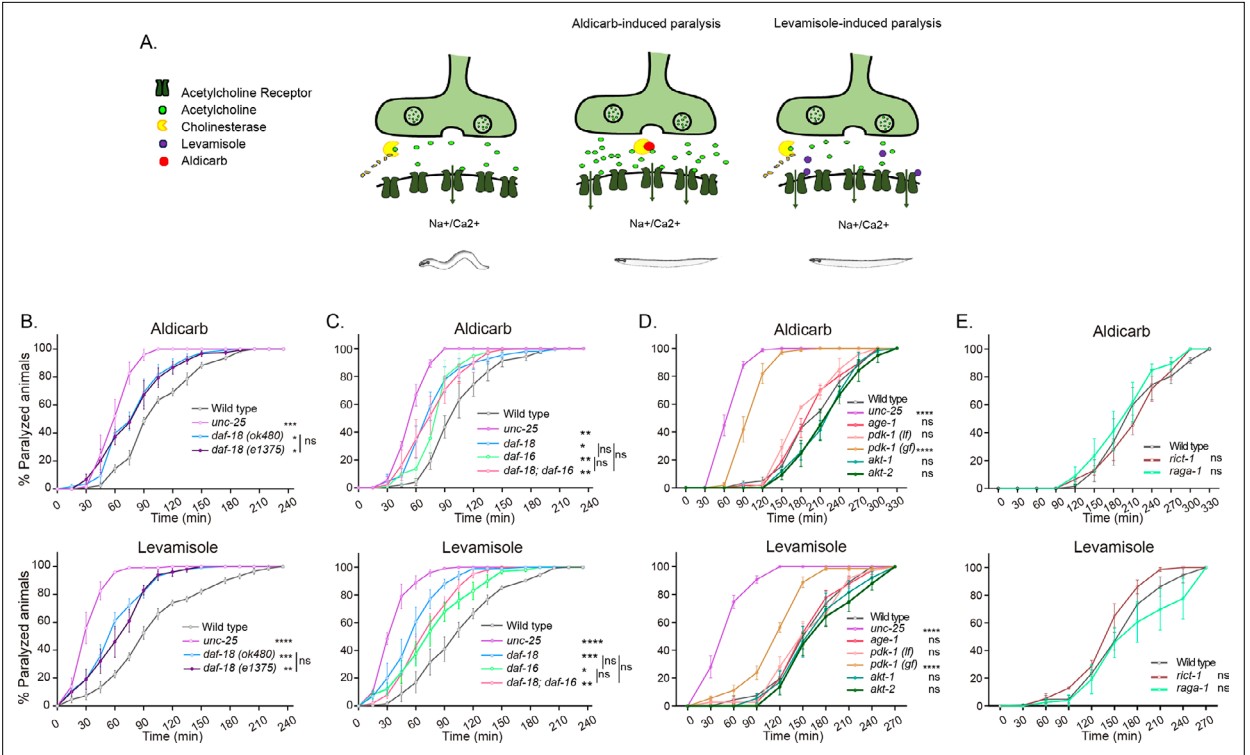

**Figure 1.** *daf-18/PTEN* mutants are hypersensitive to cholinergic drugs. (**A**) Schematic representation of paralysis induced by aldicarb and levamisole. Aldicarb acts by inhibiting acetylcholinesterase, leading to an accumulation of acetylcholine at the neuromuscular junction, resulting in continuous stimulation of muscles and worm paralysis. Levamisole functions as an agonist at nicotinic acetylcholine receptors, causing prolonged depolarization and, also, muscle paralysis. (**B–E**) Quantification of paralysis induced by aldicarb (top) and levamisole (bottom). The assays were performed in nematode growth media (NGM) plates containing 2 mM aldicarb or 0.5 mM levamisole. Strains tested: N2 (wild-type) (**B–E**), CB1375 *daf-18(e1375)* (**B**), OAR144 *daf-18(ok480)* (**B, C**), GR1310 *akt-1(mg144)* (**D**), TJ1052 *age-1(hx546)* (**D**), VC204 *akt-2(ok393)* (**D**), VC222 *raga-1(ok386)* (**E**), and KQ1366 *rict-1(ft7)* (**E**). All of these strains carry loss-of-function mutations. Furthermore, the strains denoted as '*pdk-1 (lf)*' and '*(gf)*' correspond to JT9609 *pdk-1(sa680)*, which possesses a loss-of-function mutation, and GR1318 *pdk-1(mg142)*, which harbors a gain-of-function mutation in the *pdk-1* gene, respectively. The strain CB156 *unc-25(e156)* was included as a strong GABA-deficient control (**B–D**). At least four independent trials for each condition were performed (n = 25–30 animals per trial). One-way analysis of variance (ANOVA) was used to test statistical differences in the area under the curve (AUC) among different strains. *Post hoc* analysis after one-way ANOVA was performed using Tukey's multiple comparisons test (**B, C**) and Dunnet´s to compare against the wild-type strain (**D, E**) (ns p > 0.05; *p ≤ 0.05; **p ≤ 0.01; ***p ≤ 0.001; ****p ≤ 0.0001).

The online version of this article includes the following figure supplement(s) for figure 1:

**Figure supplement 1.** *daf-18/PTEN* is the negative modulator of the phosphatidylinositol 3-phosphate kinase (PI3K)/AKT pathway.

*pdk-1* activity is linked to hyperphosphorylation and inactivation of DAF-16/FOXO (*Figure 1—figure supplement 1*), these results support the hypothesis that low activity of DAF-16/FOXO leads to hypersensitivity to these drugs.

In vertebrates, alterations in PTEN activity have been largely shown to impact neuronal development and function by affecting the mechanistic Target Of Rapamycin (mTOR) pathway (*Skelton et al., 2020*). Consequently, we analyzed whether mutations in components of the *C. elegans* TOR complexes (CeTORC) would lead to significant changes in sensitivity to aldicarb and levamisole. Our findings indicate that neither animals with loss of the essential TORC1 component *raga-1*/RagA nor animals with a loss of function in the essential TORC2 component *rict-1*/Rictor exhibited significant alterations in sensitivity to cholinergic drugs compared to wild-type animals (*Figure 1E*). This suggests that the CeTOR pathway is not involved in *daf-18/PTEN* pharmacological phenotypes.

## *daf-18/PTEN* and *daf-16/FOXO* mutants show phenotypes indicative of GABAergic deficiency

In *C. elegans*, the body wall muscles receive cholinergic innervation, which induces contraction, and GABAergic innervation, which leads to relaxation. The contralateral activity of cholinergic and

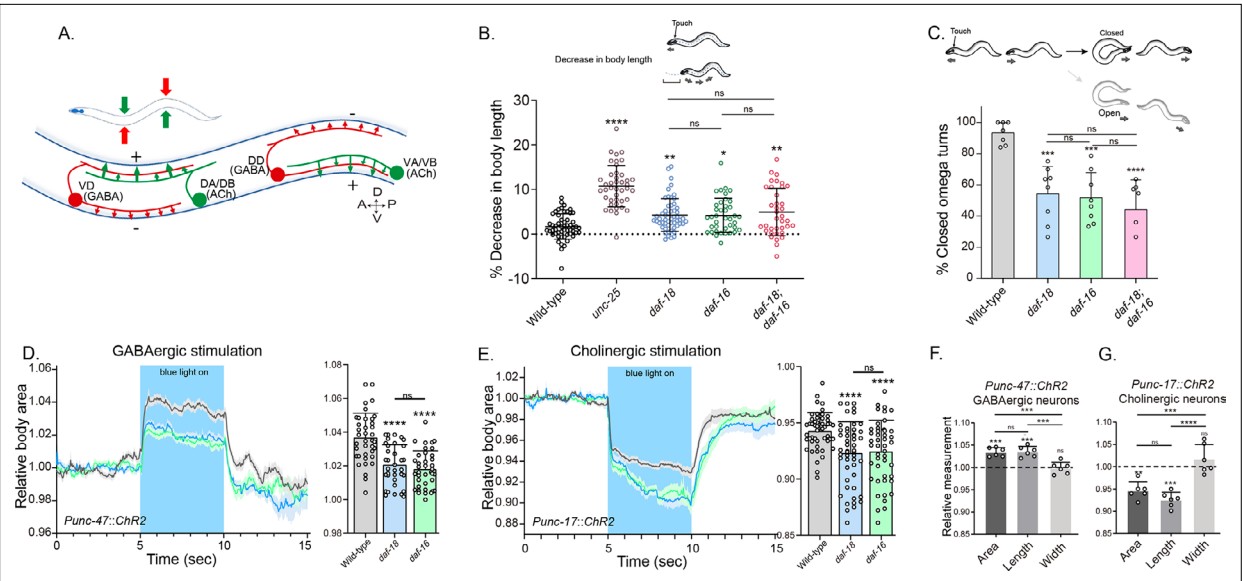

**Figure 2.** *daf-18/PTEN* mutants exhibit phenotypes typical of GABA-deficient animals. (**A**) Schematic of *C. elegans* adult neuromuscular circuit. Red indicates GABAergic motor neurons (DD/VD) and green indicates cholinergic motor neurons (VA/VB and DA/DB). The VA and VB cholinergic motor neurons send synaptic inputs to the ventral body wall muscles and the DD GABAergic motor neurons. The release of ACh from VA/VB neurons leads to the contraction of the ventral body wall muscles and the activation of DD GABAergic motor neurons that release GABA on the opposite side of the worm, causing relaxation of the dorsal body wall muscles. Conversely, activation of the DA and DB cholinergic motor neurons produces contraction of the dorsal body wall muscles and activates the VD GABAergic motor neurons. The VD GABAergic motor neurons release GABA, causing relaxation of the ventral body wall muscles, and thus contralateral inhibition. (**B**) Quantification of body shortening in response to anterior touch. Data are represented as mean ± standard deviation (SD). $n$ = 50–70 animals per genotype distributed across four independent experiments. Kruskal–Wallis analysis with Dunn's post-test for multiple comparisons was performed. (**C**) (Top) Scheme of *C. elegans* escape response in nematode growth media (NGM) agar. After eliciting the escape response by an anterior gentle touch, the omega turns were classified as closed (head and tail are in contact) or open (no contact between head and tail). (Bottom) Quantification of % closed omega turns/total omega turns. At least six independent trials for each condition were performed ($n$ = 20–25 animals per genotype/trial). Data are represented as mean ± SD. One-way analysis of variance (ANOVA) with Tukey's post-test for multiple comparisons was performed. (**D-E**) Light-evoked elongation/contraction of animals expressing Channelrhodopsin (ChR2) in GABAergic (**D**) and cholinergic (**E**) motorneurons. Animals were filmed before, during, and after a 5-s pulse of 470 nm light stimulus (15 frames/s). The body area in each frame was automatically tracked using a custom FIJI-ImageJ macro. The averaged area of each animal during the first 125 frames (0–5 s) established a baseline for normalizing light-induced body area changes. To compare the changes induced by optogenetic activity between different strains, the body area measurements for each animal were averaged from second 6 (1 s after the blue light was turned on) to second 9 (1 s before the light was turned off). These mean ± SD values are depicted in the bar graph shown to the right of each trace representation ($n$ = 40–55 animals per genotype). Tukey's multiple comparisons method following one-way ANOVA was performed for D, while Dunn's multiple comparisons test after Kruskal–Wallis analysis was used in E. (**F, G**) Manual Measurement of body length and width upon optogenetic stimulation of GABAergic (**F**) and cholinergic (**G**) neurons. At the 2.5 s time point of light stimulation, we manually measured both the width and length of multiple animals and compared these measurements with the corresponding areas obtained from automated analysis (see Materials and methods). The *Y*-axis represents the ratio between measurements taken before and after blue light illumination. We performed an ANOVA with Tukey's post hoc analysis for multiple comparisons using the normalized value (1.000) as the baseline. The area and length show significant differences compared to the measurements prior to illumination and do not differ significantly from each other indicating they vary together (increasing for *unc-47::ChR2* and decreasing for *unc-17::ChR2* animals upon blue light stimulation). In contrast, the width values are not statistically different from the pre-stimulation measurements ($p$ = 0.505 for *unc-47::ChR2*; $p$ = 0.996 for *unc-17::ChR2*). This suggests that the changes in area detected by our automated method are due to variations in length. The six animals analyzed to validate the automated measurement were randomly selected from the optogenetic experiments. Data are shown as mean ± SD (ns $p$ > 0.05; *$p \leq$ 0.05; **$p \leq$ 0.01; ***$p \leq$ 0.001; ****$p \leq$ 0.0001).

GABAergic neurons facilitates the characteristic undulatory movement of the animal (*Figure 2A*). Hypersensitivity to cholinergic drugs has long been observed in worms where GABAergic signaling is deficient (*Huang et al., 2019b*; *Vashlishan et al., 2008*; *Figure 1*). In mutants with severe deficits in GABA transmission, prodding induces a bilateral contraction of the body wall muscles that shortens the body (shrinker phenotype) (*McIntire et al., 1993*). When *daf-18/PTEN* mutants are touched, there is a slight but significant shortening in body length (*Figure 2B*). As expected, this shortening is not as noticeable as in animals with a complete deficit in GABAergic signaling, such as mutants in *unc-25*. Similar to *daf-18/PTEN* mutants, *daf-16/FOXO* animals also exhibit a mild decrease in body length

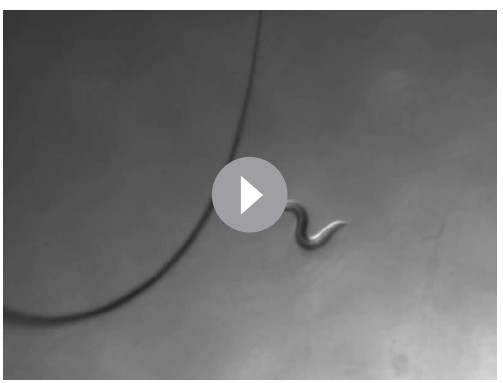

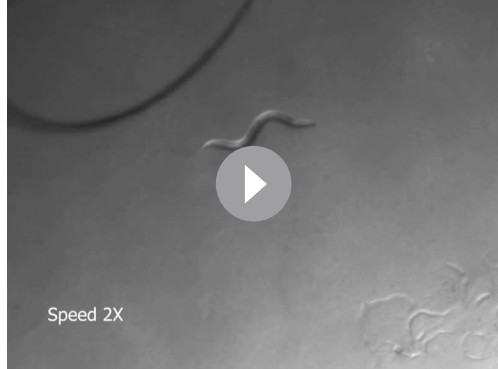

**Video 1.** A wild-type animal performing a full omega turn in response to anterior touch. After a prolonged backward movement, the animal makes a pronounced ventral turn, with its head making contact with and gliding along the ventral side of its body, before resuming forward movement in the direction opposite to its original path.

https://elifesciences.org/articles/94520/figures#video1

**Video 2.** A *daf-18* mutant animal does not execute a complete omega turn in response to an anterior touch. Following a long reversal, the animal's head fails to touch the ventral side of the body.

https://elifesciences.org/articles/94520/figures#video2

after prodding. Consistent with our aldicarb and levamisole results, there are no significant differences in body shortening between *daf-18;daf-16* double mutants and the corresponding single mutants (*Figure 2B*), further supporting the notion that both genes act in the same pathway to impact neuromuscular signaling.

We also analyzed other behaviors that require a concerted activity of GABAergic and cholinergic systems, such as the omega turns during the escape response (*Donnelly et al., 2013*). In *C. elegans* the escape response can be induced by a gentle touch on the head and involves a backward movement that is usually followed by a sharp omega turn and a 180° change in its direction of locomotion (*Donnelly et al., 2013*; *Video 1*). The execution of the omega turn involves a hypercontraction of the ventral muscles and relaxation of the dorsal muscles, allowing the animal to make a sharp turn, where the animal's head slides on the ventral side of the body (closed omega turn), and resumes locomotion in the opposite direction (*Video 1*). In response to anterior touch, the vast majority of wild-type worms make a closed omega turn (*Donnelly et al., 2013*; *Pirri and Alkema, 2012*; *Figure 2C*). Ventral muscle contraction is triggered by cholinergic motor neurons (VA and VB neurons) that synapse onto ventral muscles, while dorsal muscle relaxation is induced by GABAergic motor neurons (DD neurons) that synapse onto dorsal muscles (*Figure 2A*; *Donnelly et al., 2013*; *Pirri and Alkema, 2012*). Ablation of DD GABAergic neurons reduces dorsal muscle relaxation, therefore preventing

the head from touching the ventral side of the body during the escape response (open omega turn) (*Donnelly et al., 2013*). In agreement with previous reports, we found that 91% of wild-type animals exert a closed omega turn within the escape response (*Figure 2C*). We observed that similar to wild-type animals, gentle anterior touch with an eyelash induces *daf-18* mutants to move backward and initiate an omega turn (*Video 2*). However, only 53% of *daf-18* mutants exhibit the typical head-to-tail contact during the omega turn (*Figure 2C* and *Video 2*). Akin to *daf-18* mutants, *daf-16/FOXO* mutants exhibited a decrease in the proportion of closed omega turns (*Figure 2C*). No additive effects were observed in the *daf-18;daf-16* double mutant, suggesting

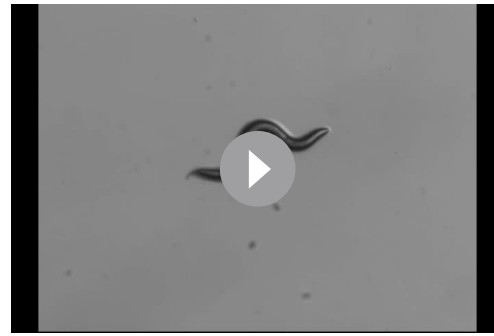

**Video 3.** Optogenetic activation of an animal expressing Channelrhodopsin (ChR2) in GABAergic neurons. Upon stimulation of the GABAergic neurons with blue light, the massive relaxation of the body wall muscles leads to an elongation of its body length.

https://elifesciences.org/articles/94520/figures#video3

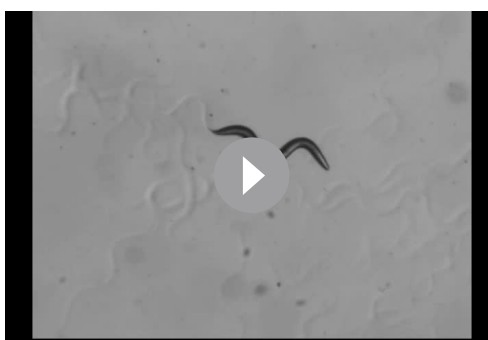

**Video 4.** Optogenetic activation of an animal expressing Channelrhodopsin (ChR2) in cholinergic neurons. Upon stimulation of the cholinergic neurons with blue light, the massive contraction of the body wall muscles leads to a shortening of its body length.
https://elifesciences.org/articles/94520/figures#video4

that the increased inactivation of *daf-16/FOXO* is primarily responsible for the defects observed in the escape response of *daf-18* mutants.

Given that our results suggest a deficit in GABAergic functionality in *daf-18/PTEN* mutants, we used optogenetics to specifically activate these neurons in mutant worms. The expression of Channelrhodopsin (ChR2) in GABAergic motor neurons (using the promoter of the *unc-47* gene, an ortholog of the vesicular GABA transporter SLC32A1) elicits a flaccid paralysis of the worms upon exposure to blue light. This obvious and robust response results in an increase in body length that can be used as a clear readout (*Hwang et al., 2016*; *Schultheis et al., 2011*; *Koopman et al., 2021*; *Figure 2D* and *Video 3*). Interestingly, we found that the elongation of the animal after the specific activation of GABAergic neurons is significantly decreased in *daf-18/PTEN* and *daf-16/FOXO* mutants compared to wild-type worms (*Figure 2D*). While these results suggest a defect in GABAergic transmission, it could also be possible that general neuronal transmission is affected. Consequently, we reciprocally activated the cholinergic motor neurons in animals expressing ChR2 under the *unc-17* promoter, a gene encodes the vesicular acetylcholine transporter (VAChT), which leads to muscle contraction and shortened body length (*Hwang et al., 2016*; *Koopman et al., 2021*; *Figure 2E* and *Video 4*). Rather than observing reduced shortening in *daf-16/FOXO* and *daf-18/PTEN* mutants, we found that cholinergic activation caused hypercontraction of these mutant animals (*Figure 2E*). Since the activation of cholinergic motor neurons not only activates muscles but also stimulates GABAergic neurons to produce counteractive muscle relaxation in the other side of the animal (*Figure 2A*), it is expected that a GABAergic deficit would lead to increased muscle contraction and body shortening upon cholinergic activation. In summary, these results strongly suggest that in *daf-18/PTEN* and *daf-16/FOXO* mutants, there is a specific functional defect in GABAergic neurons, while excitatory neurons do not appear to be affected.

## Disruption of *daf-18/PTEN* alters commissural trajectories in GABAergic motor neurons

Since our previous results imply perturbations of neuromuscular transmission, we explored the morphology of *C. elegans* motor neurons. The cell bodies of both cholinergic (A- and B-type) and GABAergic (D-type) motor neurons that innervate body wall muscles are located in the ventral nerve cord, and a subset extends single process commissures to the dorsal nerve cord (DNC) (*Altun and Hall, 2011*; *Figure 3—figure supplement 1*). The commissures have proved useful for studying defects in motor neuron development or maintenance (*de Cáceres et al., 2012*; *Oliver et al., 2019*). We analyzed the morphology of GABAergic motor neurons in L4 animals expressing *mCherry* under the control of the *unc-47* promoter (*Byrne et al., 2016*). We found that *daf-18/PTEN* mutants exhibit a higher frequency of commissure flaws, including guidance defects, ectopic branching, and commissures that fail to reach the dorsal cord (*Figure 3B, C*). In contrast to our findings in GABAergic neurons, we observed no obvious differences in the frequency of commissure defects when we compared cholinergic motor neurons in control and *daf-18/PTEN* animals (*Figure 3A*, *Figure 3—figure supplement 1*).

GABAergic motor neurons can be classified based on the muscles they innervate: those that innervate the dorsal muscles are called DDs, while those that innervate the ventral muscles are called VDs. Both types of D neurons send commissures to the DNC (*Sulston and White, 1980*; *White et al., 1986*). We found defects in various commissures, some of which correspond to VD and others to DD neurons (*Figure 3—figure supplement 2*). We also analyzed GABAergic commissures at the beginning (1 hr post-hatching) of the first larval stage (L1), when only the six DD neurons are formed

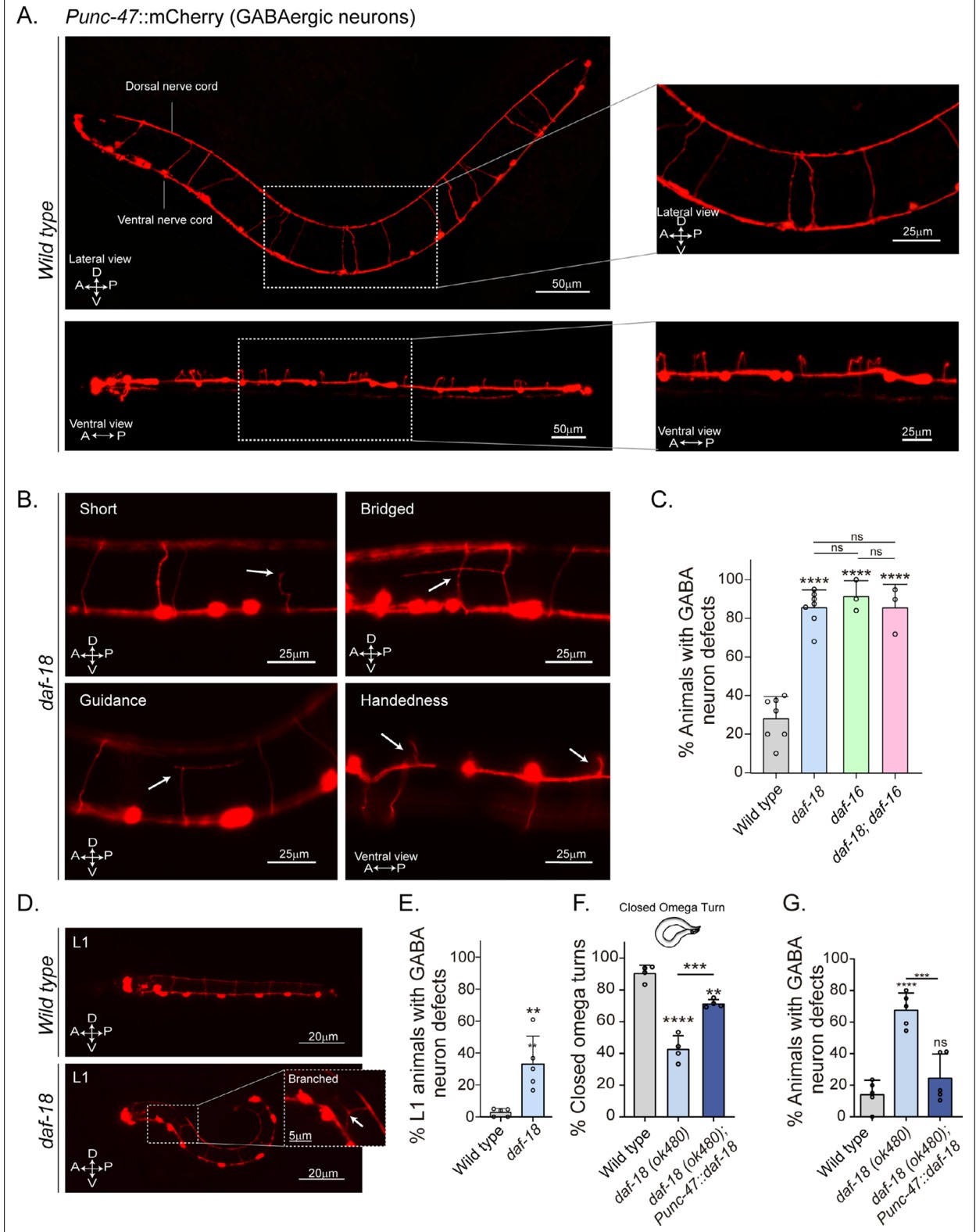

**Figure 3.** *daf-18/PTEN* mutants show neurodevelopmental defects in GABAergic motor neurons. (**A**) Representative images of wild-type animals expressing *mCherry* in the GABAergic motor neurons are shown laterally (top) and ventrally (bottom). In the insets, commissures are depicted at a higher resolution. Note that in the ventral view, all the processes travel through the right side of the animal's body. (**B**) Representative images of commissure defects observed in *daf-18 (ok480)* mutants (arrows). The defects shown are: Short, commissure length less than half of nematode width;

*Figure 3 continued on next page*

*Figure 3 continued*

Bridged, neighboring commissures linked by a neurite; Guidance, commissures that do not reach dorsal nerve cord; and Handedness, commissure running along the opposite side of the animal's body. (C) Quantification of GABAergic system defects. Each bar represents the mean ± standard deviation (SD). One-way analysis of variance (ANOVA) and Tukey's multiple comparisons test were used for statistics (ns p > 0.05; ****p ≤ 0.0001). At least three independent trials for each condition were performed (*n*: 20–25 animals per genotype/trial). (D) Representative image of L1 animals (1 hr post-thatch) expressing *Punc-47::mCherry* in wild-type (top) and *daf-18(ok480)* mutant (bottom) backgrounds. In this larval stage, only six GABAergic DD motor neurons are born. The inset shows a typical defective (branched) commissure. (E) Quantification of GABAergic system defects in L1s. Each bar represents the mean ± SD. Two-tailed unpaired Student's *t*-test (**p ≤ 0.01). At least five independent trials for each condition were performed (*n*: ~20 animals per genotype/trial). (F, G) Quantification of closed omega turns/total omega turns and commissure defects in GABAergic neurons of animals expressing *daf-18/PTEN* solely in GABAergic neurons. One-way ANOVA and Tukey's multiple comparisons test were used for statistics (ns p > 0.05; **p ≤ 0.01; ***p ≤ 0.001; ****p ≤ 0.0001). At least four independent trials for each condition were performed (*n*: 15–20 animals per genotype/trial). A – anterior; P – posterior; D – dorsal; V – ventral.

The online version of this article includes the following figure supplement(s) for figure 3:

**Figure supplement 1.** *daf-18/PTEN* mutations do not affect excitatory cholinergic motor-neuron morphology.

**Figure supplement 2.** *daf-18/PTEN* deficiencies affect DDs and VDs GABAergic neurons.

**Figure supplement 3.** DD-GABAergic neurons show defects in recently hatched L1 animals of *daf-18/PTEN* mutants.

(*Sulston and White, 1980*; *Hallam and Jin, 1998*; *Mulcahy et al., 2022*). We found that in this early larval stage, *daf-18/PTEN* mutants exhibit defects in the GABAergic commissures (*Figure 3D, E* and *Figure 3—figure supplement 3*). While the DD neurons are born and mostly develop embryonically, limited post-embryonic axonal outgrowth has been observed in these neurons (*Mulcahy et al., 2022*). We did not find an increase in the number of errors in larvae 5–6 hr post-hatching compared to recently hatched larvae (*Figure 3—figure supplement 3*), indicating that deficiencies in DD neurons in *daf-18/PTEN* mutants mainly occur during their embryonic development. In contrast, we observed that the prevalence of errors increases significantly at the L4 stage (*Figure 3—figure supplement 3*). The greater number of defects in L4s likely arises from defects in the VDs, which are born post-embryonically between the mid-L1 larval stage and the L2 stage, and add to the defects already present in the DDs. Taken together, these observations suggest that reduced DAF-18/PTEN activity affects the neurodevelopment of the GABAergic motor system. Since the transcription factor DAF-16/FOXO is one of the main targets of DAF-18/PTEN signaling, we analyzed the morphology of GABAergic motor neurons in *daf-16/FOXO* null mutants. These animals also exhibit an increased number of defects in GABAergic commissures compared to the wild type (*Figure 3C*).

Given that *daf-18* is ubiquitously expressed in all tissues (*Ogg and Ruvkun, 1998*), we asked whether DAF-18/PTEN acts autonomously in GABAergic neurons to ensure proper development. We found that specific *daf-18/PTEN* expression in GABAergic neurons increased the proportion of closed omega turns in *daf-18/PTEN* null mutants (*Figure 3F*). In addition, the morphological defects in GABAergic commissures were significantly reduced (*Figure 3G*), suggesting that DAF-18/PTEN acts autonomously in GABAergic motor neurons to regulate their development.

Our findings demonstrate that mutations in *daf-18/PTEN* and *daf-16/FOXO* result in developmental defects in GABAergic neurons, leading to altered morphology and function while leaving cholinergic motor neurons unaffected. Our experiments strongly suggest that these defects in the inhibitory transmission arise from the hyperactivation of the PI3K pathway, along with subsequent DAF-16/FOXO inhibition, in GABAergic neurons of *daf-18/PTEN* mutants.

## A diet enriched with the ketone body βHB ameliorates defects in *daf-18/PTEN* mutants but not in *daf-16/FOXO* mutants

Mutations in *PTEN* are linked with ASDs (*Rademacher and Eickholt, 2019*). KGDs, which force the endogenous production of Ketone Bodies (KBs), have proved to be effective for the treatment of neurological disorders associated with E/I imbalances, such as epilepsy and, more recently, ASD (*Neal et al., 2008*; *Lambrechts et al., 2017*; *Li et al., 2021*). It has been shown that the KB βHB induces DAF-16/FOXO activity (*Edwards et al., 2014*). Therefore, we asked whether it is possible to improve the observed phenotypes by modulating the activity of DAF-16/FOXO with βHB. We first evaluated the expression of *sod-3*, which codes for a superoxide dismutase and is a DAF-16/FOXO transcriptional target gene (*De Rosa et al., 2019*). We used a strain expressing a GFP transcriptional reporter for *sod-3* and determined fluorescence intensity upon dietary supplementation of βHB. Consistent

with previous reports, the levels of SOD-3::GFP are reduced in *daf-18/PTEN* and *daf-16/FOXO* mutant strains. Furthermore, we observed that βHB (20 mM) induces the expression of *sod-3* in *daf-18/PTEN* but not in *daf-16/FOXO* mutants (*Figure 4—figure supplement 1*). Importantly, we did not detect increased *sod-3* expression in *daf-18; daf-16* double deficient animals, strongly suggesting that βHB induces *sod-3* expression in *daf-18/PTEN* mutants through the transcription factor *daf-16/FOXO* (*Figure 4—figure supplement 1*).

Next, we evaluated behavioral phenotypes and GABAergic neuronal morphology of animals that were raised on an *Escherichia coli* diet supplemented with 20 mM βHB throughout development, from egg laying until the time of the assay (typically late L4s or young adults). We found that βHB supplementation significantly reduced the hypersensitivity of *daf-18/PTEN* mutants to the cholinergic drugs aldicarb and levamisole (*Figure 4A, B*). Moreover, βHB supplementation rescued the post-prodding shortening in *daf-18/PTEN* mutants (*Figure 4C*). Accordingly, we found that *daf-18/PTEN* mutants showed a significant increase in the proportion of closed omega turns during their escape response compared to the naive condition (*Figure 4D*). In contrast, βHB exposure does not change the number of closed omega turns in *daf-16/FOXO* null mutants or the double null mutant *daf-18; daf-16* (*Figure 4D*).

We subsequently analyzed the changes in body length induced by optogenetic activation of both GABAergic and cholinergic neurons in animals exposed to a diet enriched with βHB. Interestingly, we found that *daf-18/PTEN* mutants exposed to βHB, but not wild-type or *daf-16/FOXO* mutant animals, exhibited increased elongation following optogenetic activation of GABAergic neurons (*Figure 4E–G*). Furthermore, we observed that the hypercontraction observed in *daf-18/PTEN* mutants after the activation of cholinergic neurons is significantly reduced in animals exposed to βHB (*Figure 4H–J*). These findings suggest that this ketone body can rebalance excitatory and inhibitory signals in the neuromuscular system of *daf-18/PTEN C. elegans* mutants.

We also evaluated the morphology of GABAergic motor neurons in *daf-18/PTEN* animals exposed to βHB. We found that βHB supplementation reduced the frequency of defects in GABAergic processes (*Figure 4K*). Consistently, βHB exposure did not significantly reduce the defects on GABAergic neurons of either *daf-16/FOXO* null mutants or *daf-18; daf-16* double mutants. Taken together, these results demonstrate that dietary βHB ameliorates the defects associated with deficient GABAergic signaling in *daf-18/PTEN* mutants.

It is noteworthy that we did not observe any improvement in either neuronal outgrowth defects in the AIY interneuron or the migration of the HSN motor neurons (*Figure 4—figure supplement 2*) in *daf-18/PTEN* mutants exposed to βHB, even though these defects were shown to depend on the reduction of DAF-16/FOXO activity (*Christensen et al., 2011*; *Kennedy et al., 2013*). AIY neurite and HSN soma migration take place during embryogenesis (*Christensen et al., 2011*; *Hedgecock et al., 1987*). It is therefore possible that βHB may not go through the impermeable chitin eggshell of the embryo, as has been reported with other drugs (*Sato et al., 2006*).

## βHB exposure during early L1 development sufficiently mitigates GABAergic system defects

In the above experiments, animals were exposed to βHB throughout development. We next asked whether there is a critical period during development where the action of βHB is required. We exposed *daf-18/PTEN* mutant animals to βHB-supplemented diets for 18-hr periods at different developmental stages (*Figure 5A*). The earliest exposure occurred during the 18 hr following egg laying, covering ex utero embryonic development and the first 8–9 hr of the L1 stage. The second exposure period encompassed the latter part of the L1 stage, the entire L2 stage, and most of the L3 stage. The third exposure spanned the latter part of the L3 stage, the entire L4 stage, and the first 6–7 hr of the adult stage (*Figure 5A*). Interestingly, we found that the earliest exposure to βHB was sufficient to increase the proportion of *daf-18/PTEN* mutant animals executing a closed omega turn during the escape response. However, when the animals were exposed to βHB at later juvenile stages, their ability to enhance the escape response of *daf-18/PTEN* mutants declined (*Figure 5B*). Moreover, exposing animals to βHB for 18 hr starting from egg laying was enough to reduce morphological defects in the GABAergic motor neurons of these mutants (*Figure 5C, D*). Interestingly, βHB has no effect on GABAergic commissures in either recently hatched L1s or L4 *daf-18/PTEN* mutants when exposure is limited to the first 9 hr after egg laying (where ex utero embryonic development occurs), possibly due

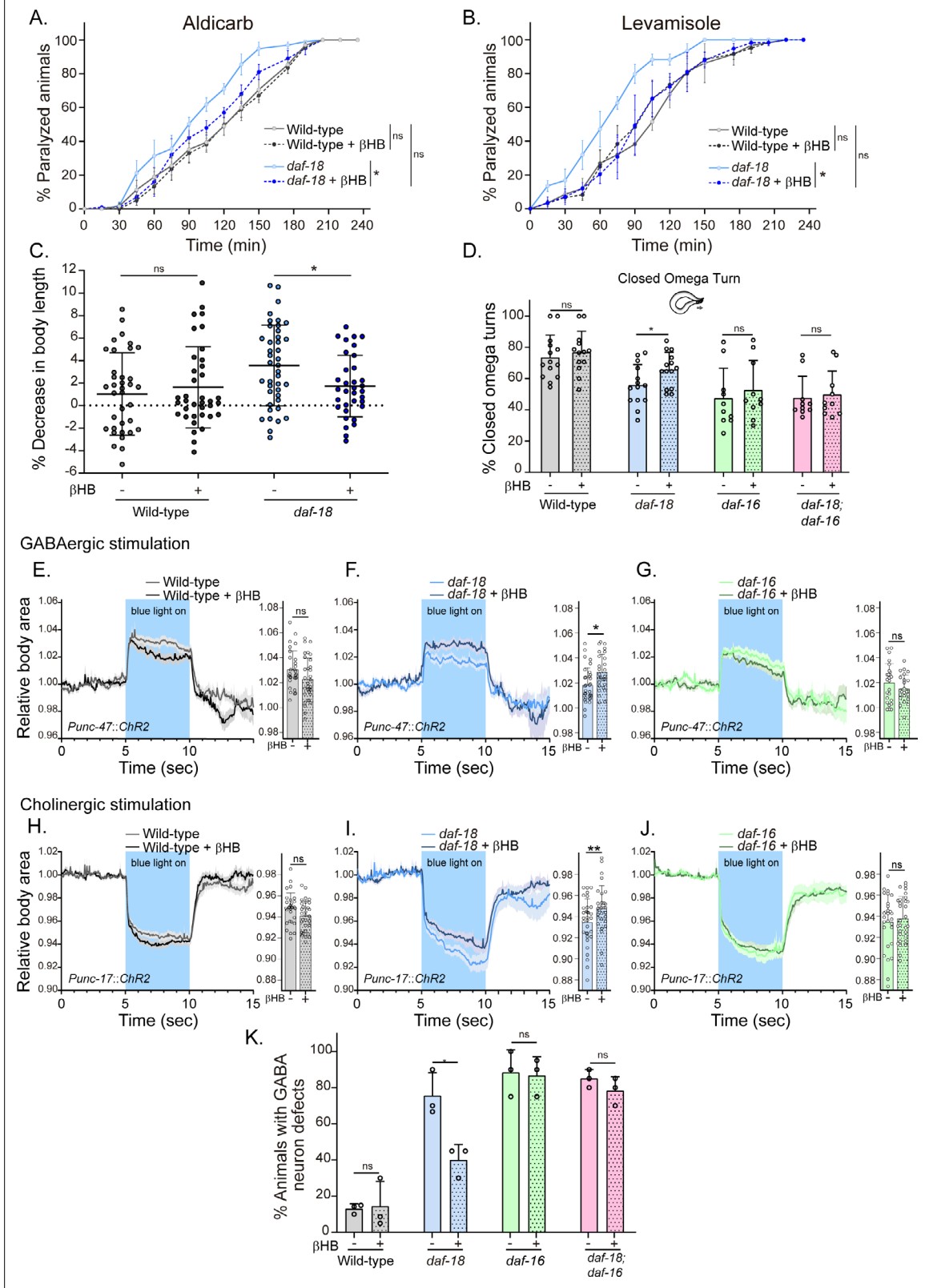

**Figure 4.** Dietary β-hydroxybutyrate (βHB) supplementation ameliorates GABAergic deficits in *daf-18/PTEN* mutants. Animals were exposed to βHB (20 mM) throughout development (from embryo to L4/young adults). (**A, B**) Quantification of paralysis induced by cholinergic drugs. At least four independent trials for each condition were performed (*n*: 20–25 animals per genotype/trial). Two-tailed unpaired Student's *t*-test (ns p > 0.05; *p ≤ 0.05; **p ≤ 0.01) was used to compare βHB treated and untreated animals. (**C**) Measurement of body length in response to anterior touch. *n* = 30–40 animals

*Figure 4 continued on next page*

*Figure 4 continued*

per genotype distributed across three independent experiments. Two-tailed unpaired Student's *t*-test was used to compare βHB treated and untreated animals (ns p > 0.05; *p ≤ 0.05). (**D**) Quantification of closed omega turns/total omega turns during the escape response. At least eight independent trials for each condition were performed (*n* = 20 animals per genotype/trial). Results are presented as mean ± standard deviation (SD). Two-tailed unpaired Student's *t*-test (ns p > 0.05; *p ≤ 0.05). Light-evoked elongation/contraction of animals expressing Channelrhodopsin (ChR2) in GABAergic (**E–G**) and cholinergic (**H–J**) motorneurons. The mean body area (mean ± SD) during 3 s of the light pulse is depicted in the bar graph shown to the right of each trace representation (see *Figure 2*) (*n* = 25–35 animals per condition). Two-tailed unpaired Student's *t*-test (ns p > 0.05; *p ≤ 0.05; **p ≤ 0.01). (**K**) Quantification of commissure defects in GABAergic neurons. Results are presented as mean ± SD. Two-tailed unpaired Student's *t*-test (ns p > 0.05; *p ≤ 0.05). At least three independent trials for each condition were performed (*n* = ~20 animals per genotype/trial).

The online version of this article includes the following figure supplement(s) for figure 4:

**Figure supplement 1.** Exposure to β-hydroxybutyrate (βHB) induces *sod-3* expression in *daf-18/PTEN*, but not in *daf-16/FOXO* mutants.

**Figure supplement 2.** β-Hydroxybutyrate (βHB) does not prevent neurodevelopmental defects in AIY and HSN neurons.

to the impermeability of the chitinous eggshell (*Figure 5D*). Thus, it is likely that βHB acts at early L1 stages to mitigate neurological GABAergic defects in *daf-18/PTEN* mutants.

Taken together, our findings demonstrate that mutations in *daf-18*, the *C. elegans* ortholog of *PTEN*, lead to defects in inhibitory GABAergic neurodevelopment without significantly affecting cholinergic excitatory signals. These deficiencies in GABAergic neurons manifest as altered neuronal morphology, hypersensitivity to cholinergic stimulation, reduced responses to optogenetic GABAergic neuronal activation, mild body shortening following touch stimuli, and deficits in the execution of the omega turn. We have determined that these impairments in GABAergic development result from reduced activity of the FOXO ortholog DAF-16 in *daf-18/PTEN* mutants. Importantly, our study's pivotal finding is that a βHB-enriched diet during early development, robustly mitigates the deleterious effects of *daf-18/PTEN* mutations in GABAergic neurons. This protective effect is critically dependent on the induction of DAF-16/FOXO by this ketone body.

## Discussion

Mutations in *daf-18/PTEN* are linked to neurodevelopmental defects from worms to mammals (***Chen et al., 2015***; ***Christensen et al., 2011***; ***Kennedy et al., 2013***). Moreover, decreased activity of PTEN produces E/I disequilibrium and the development of seizures in mice (***van Diepen and Eickholt, 2008***). The mechanisms underlying this imbalance are not clear. Our results demonstrate that reduced DAF-18/PTEN activity in *C. elegans* generates guidance defects, abnormal branching, incomplete commissural outgrowth, and deficient function of inhibitory GABAergic neurons, without affecting the excitatory cholinergic neurons.

*daf-18/PTEN*-deficient mutants have a shorter lifespan (***Mihaylova et al., 1999***). One possibility is that the defects in GABAergic processes are due to neurodegeneration associated with premature aging rather than developmental flaws. However, this idea is unlikely given that the neuronal defects in DD neurons are already evident at the early L1 stage. In *C. elegans*, DD GABAergic motor neurons undergo rearrangements during the L1–L2 stages (***Hallam and Jin, 1998***; ***Mulcahy et al., 2022***). In newly hatched L1 larvae, each DD motor neuron innervates ventral muscles and extends a commissure to the DNC to receive synaptic inputs from cholinergic DA and DB neurons (***Hallam and Jin, 1998***; ***Mulcahy et al., 2022***; ***Cuentas-Condori and Miller Rd, 2020***). In adults, the DD commissure morphology is maintained, but the synaptic output shifts to dorsal muscles, and input is provided by VA and VB cholinergic motor neurons in the ventral nerve cord (***Cuentas-Condori and Miller Rd, 2020***). A plausible possibility is that this remodeling process makes DD neurons specifically sensitive to PI3K pathway activity. However, since the DD commissures are defective already at the very early L1 stage (prior to rewiring) and similar defects are observed in VD neurons, which are born post-embryonically between late L1 and L2 stages and do not undergo this remodeling (***Sulston and White, 1980***; ***Mulcahy et al., 2022***; ***Cuentas-Condori and Miller Rd, 2020***), we can infer that the deficits caused by *daf-18/PTEN* deficiency affect the development of the entire GABAergic system, independently of the synaptic rearrangement of DD neurons.

Strikingly, cholinergic neurons have no noticeable morphological or functional defects in *daf-18/PTEN* mutants. Loss-of-function mutants in the neuronal integrin *ina-1*, ortholog of human *ITGA6*, affect the guidance of GABAergic commissures, without affecting cholinergic neurons (***Oliver et al.,***

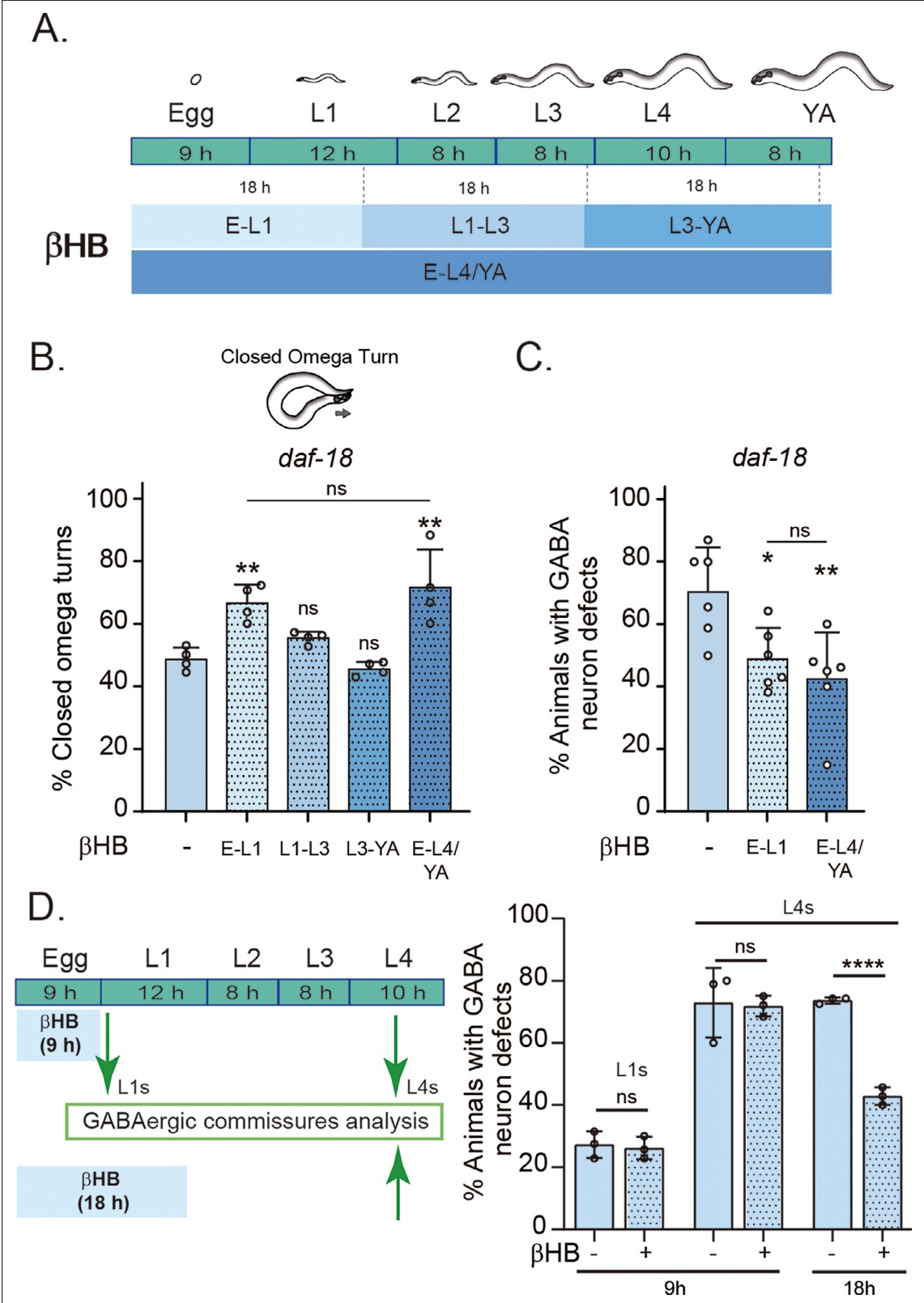

**Figure 5.** Early developmental stages are critical for β-hydroxybutyrate (βHB) modulation of GABAergic signaling. (**A**) Animals were exposed to βHB-enriched diet for 18 hr periods at different developmental stages: (1) E-L1 covered ex utero embryonic development (~9 hr) and the first 8–9 hr of the L1 stage; (2) L1–L3 covered the latter part of the L1 stage (~3–4 hr), the entire L2 stage (~8 hr), and most of the L3 stage (~6–7 hr); (3) L3-YA (Young Adult) spanned the latter part of the L3 stage (~1–2 hr), the entire L4 stage (~10 hr), and the first 6–7 hr as adults, and (4) E-L4/YA implies exposure throughout

*Figure 5 continued*

development (from embryo to Young Adult). Quantification of closed omega turns/total omega turns in *daf-18/PTEN* (**B**) and GABAergic commissure defects (**C**) in *daf-18/PTEN* mutants exposed to βHB at different developmental intervals. Four and six independent trials for each condition were performed in B and C, respectively (*n* = 20–25 animals per genotype/trial). Results are presented as mean ± standard deviation (SD). One-way analysis of variance (ANOVA) with Tukey's post-test for multiple comparisons was performed (ns p > 0.05; *p ≤ 0.05; **p ≤ 0.01). (**D**) βHB does not prevent neurodevelopmental defects in GABAergic neurons when applied exclusively during ex utero embryonic development. Quantification of GABAergic commissure defects in L4-stage of *daf-18/PTEN* mutant animals exposed to βHB during the first 9 hr post-egg laying (just before hatching) and 18 hr post-egg laying. The animals were then transferred to control plates without βHB and maintained until GABAergic commissures analysis. Scoring was performed in 0.5–1 hr post-hatching (early L1 larva) and L4 animals (green arrows). A two-tailed unpaired Student's *t*-test was used for statistical analysis. Data represent three independent trials with at least 20 worms per trial. Results are presented as mean ± SD. (ns P>0.05; **** P≤0.0001).

*2019*). Similar to *PTEN*, mutations in neuronal integrins have been linked to neurodevelopmental defects (*Wu and Reddy, 2012*). Interestingly, the PI3K/Akt/FOXO pathway and integrin signaling are interrelated in mammals (*Moreno-Layseca and Streuli, 2014*). This observation opens the possibility that one of the mechanisms by which *daf-18/PTEN* mutants have defects in GABAergic neurodevelopment involves integrin expression and/or function. Interestingly, mutations in *eel-1*, the *C. elegans* ortholog of *HUWE1*, or in subunits of the Anaphase-Promoting Complex, lead to developmental and functional alterations in GABAergic neurons but not in cholinergic neurons (*Opperman et al., 2017*; *Kowalski et al., 2014*), despite their expression in both neuronal types. This observation suggests the existence of compensatory or redundant mechanisms in cholinergic neurons that may not be present in GABAergic neurons.

In mammals, defects in *PTEN* mutants have been typically related to altered function of the mTOR pathway (*Kwon et al., 2006*; *Cupolillo et al., 2016*; *Huang et al., 2019a*). However, our results suggest that, in the *C. elegans* neuromuscular system, decreased activity of DAF-18/PTEN affects GABAergic development due to a downregulation of DAF-16/FOXO transcription factor activity. The FOXO family of transcription factors is conserved throughout the animal kingdom (*Arden, 2008*). There is increasing evidence demonstrating the key role of this transcription factors family in neurodevelopment (*Santo and Paik, 2018*; *McLaughlin and Broihier, 2018*; *de la Torre-Ubieta et al., 2010*). Downregulation of FOXO activity early in development reproduces neuropathological features found in ASD patients, that is, increased brain size and cortical thickness (*Khundrakpam et al., 2017*; *Paik et al., 2009*). The autonomic activity of DAF-18/PTEN and DAF-16/FOXO coordinates axonal outgrowth in *C. elegans* AIY interneurons and rat cerebellar granule neurons (*Christensen et al., 2011*). On the other hand, DAF-18/PTEN and DAF-16/FOXO in the hypodermis control neuronal migration of the HSN neuron during development (*Kennedy et al., 2013*). Our rescue experiments strongly suggest that the PI3K/Akt/DAF-16 pathway modulates the development of GABAergic motor neurons by acting autonomously in these cells. Noteworthy, autonomic DAF-16/FOXO activity in GABAergic motor neurons is also key for axonal growth during regeneration (*Byrne et al., 2014*). These results further emphasize the importance of DAF-16/FOXO in neuronal development and axonal growth.

In many patients suffering from epilepsy, KGDs can control seizures (*Neal et al., 2008*; *Lambrechts et al., 2017*). Furthermore, they can reduce behavioral abnormalities in individuals with ASD (*Li et al., 2021*). While the mechanisms underlying the clinical effects of KGDs remain unclear, it has been shown that these diets correlate with increased GABA signaling (*Cantello et al., 2007*; *Calderón et al., 2017*; *Yudkoff et al., 2008*). We demonstrate here that dietary supplementation of the ketone body βHB ameliorates morphological and functional defects in GABAergic motor neurons of *daf-18/PTEN* mutants. Although ketone bodies were historically viewed as simple carriers of energy to peripheral tissues during prolonged fasting or exercise, our findings confirm more recent reports showing that βHB also possesses a variety of important signaling functions (*Newman and Verdin, 2017*). We can hypothesize several distinct, non-mutually exclusive models by which βHB can induce DAF-16/FOXO-dependent signaling. βHB directly inhibits mammalian histone deacetylases HDAC1 and HDAC2, increasing histone acetylation at the *FOXO3a* promoter and inducing the expression of this gene (*Shimazu et al., 2013*). HDAC1 and HDAC2 play an important role as redundant regulators of neuronal development (*Park et al., 2022*). Interestingly, in *C. elegans* βHB inhibits the class I HDACs to extend worm lifespan in a DAF-16/FOXO-dependent manner (*Edwards et al., 2014*). Therefore, βHB-mediated HDAC

inhibition may upregulate transcription of DAF-16/FOXO counterbalancing hyperactivation of the PI3K pathway in *daf-18/PTEN* mutants. Another potential mechanism for the effect of βHB involves the inhibition of the insulin signaling pathway. In mammals, the administration of βHB downregulates the insulin signaling in muscle (*Yamada et al., 2010*). Moreover, several reports have shown that βHB administration reduces phosphorylation and activity of Akt/protein kinase downstream of the insulin receptor (*Kim et al., 2019*; *McDaniel et al., 2011*). In *C. elegans*, inhibition of AKT-1 activates DAF-16/FOXO (*Hertweck et al., 2004*). Although understanding the mechanism behind βHB's action will require further studies, our results demonstrate that this ketone body positively modulates DAF-16/FOXO during neuronal development.

Multiple reports, from *C. elegans* to mammals, suggest that there is a sensitive period, typically early in development, where pharmacological or genetic interventions are more effective in ameliorating the consequences of neurodevelopmental defects (*Meredith, 2015*). However, recent evidence shows that phenotypes associated with certain neurodevelopmental defects can be ameliorated by interventions during adulthood (*Kepler et al., 2022*). Our results show that βHB can ameliorate the phenotypic defects of *daf-18/PTEN* mutants only when exposure occurs during an early critical period. The inefficacy of βHB at later stages suggests that the role of DAF-16/FOXO in the maintenance of GABAergic neurons is not as relevant as its role in development.

Our experiments do not allow us to distinguish whether the effect of βHB is preventive, reversive, or both. Our results suggest that the improvement is not due to prevention in DDs because the defects are present in newly hatched larvae regardless of the presence or absence of βHB, and DD post-embryonic growth does not add new errors. Unlike in early L1 stages, the protective effect of βHB becomes evident when analyzing the commissures of L4 animals. In this late larval stage, not only the DDs but also the VD neurons are present. This leads us to speculate that βHB may have a preventive action on the neurodevelopment of VD neurons. We also cannot rule out that this improvement may be due, at least partially, to a reversal of defects in DD neurons. It is intriguing how exposure to βHB during early L1 could ameliorate defects in neurons that mainly emerge in late L1s (VDs). We can hypothesize that residual βHB or a metabolite from the previous exposure may prevent these defects in VD neurons. βHB, in particular, has been shown to generate long-lasting effects through epigenetic modifications (*He et al., 2023*). Further investigations are needed to elucidate the underlying fundamental mechanisms regarding the ameliorating effects of βHB supplementation on deficits in GABAergic neurodevelopment associated with mutations in *daf-18/PTEN*.

Across the animal kingdom, food signals increase insulin levels leading to the activation of Akt/PI3K pathway (*Klöckener et al., 2011*; *Britton et al., 2002*; *Kaplan et al., 2018*). In *C. elegans*, the L1 larval stage is particularly sensitive to nutritional status. *C. elegans* adjusts its development based on food availability, potentially arresting in L1 in the absence of food (*Baugh, 2013*). Strong loss-of-function alleles in the insulin signaling pathway exhibit constitutive L1 arrest (*Gems et al., 1998*), highlighting the critical importance of this pathway during this larval stage. Hence, it is not surprising that dietary interventions targeting the PI3K pathway at these critical early L1 stages can modulate developmental processes. Our pharmacological experiments showed that mutants associated with an exacerbation of the PI3K pathway, which typically inhibits the nuclear translocation and activity of the transcription factor DAF-16/FOXO, lead to E/I imbalances that manifest as hypersensitivity to cholinergic drugs. We demonstrated that these imbalances arise from defects that occur specifically in the neurodevelopment of GABAergic motor neurons. Interestingly, mutants inhibiting the PI3K pathway do not show differences in their sensitivity to cholinergic drugs compared to wild-type animals. This observation can be explained by a critical period during neurodevelopment when the Insulin/Akt/PI3K pathway must be maintained at very low activity (or even deactivated). These low activity levels of the Insulin/Akt/PI3K pathway would allow for a high level of DAF-16/FOXO activity, which, according to our results, appears to be key for the proper development of GABAergic neurons.

The fine regulation of insulin and insulin-like signaling during early development is a conserved process in animals (*Baker et al., 1993*; *Brogiolo et al., 2001*). In mammals, for instance, conditions like Gestational Diabetes Mellitus, characterized by fetal hyperinsulinemia and high levels of IGF-1 (*Matuszek et al., 2011*), are associated with neurodevelopmental defects (*Li et al., 2016*). Our results lead to the intriguing idea that dietary interventions that increase DAF-16/FOXO activity, such as βHB supplementation, could constitute a potential therapeutic strategy for these pathologies. Future studies using mammalian models are crucial to shed light on the potential of this hypothesis.

## Materials and methods

### *C. elegans* culture and maintenance

All *C. elegans* strains were grown at room temperature (22°C) on nematode growth media (NGM) agar plates with OP50 *E. coli* as a food source. The wild-type reference strain used in this study is N2 Bristol. Some of the strains were obtained through the *Caenorhabditis* Genetics Center (CGC, University of Minnesota). Worm population density was maintained low throughout their development and during the assays. All experiments were conducted on age-synchronized animals. This was achieved by placing gravid worms on NGM plates and removing them after 2 hr. The assays were performed on the animals hatched from the eggs laid in these 2 hr.

Transgenic strains were generated by microinjection of plasmid DNA containing the construct *Punc-47::daf-18cDNA* (kindly provided by Alexandra Byrne, UMASS Chan Medical School) at 20 ng/µl into the germ line of (*daf-18 (ok480); lin-15 (n765ts)*) double mutants with the co-injection marker *lin-15* rescuing plasmid pL15EK (80 ng/µl). At least three independent transgenic lines were obtained. Data are shown from a single representative line.

The strains used in this manuscript were:

CB156 *unc-25(e156)* III
MT6201 *unc-47(n2409)* III
CB1375 *daf-18(e1375)* IV
OAR144 *daf-18(ok480)* IV
GR1307 *daf-16(mgdf50)* I
OAR115 *daf-16(mgDf50)* I; *daf-18(ok480)* IV
OAR161 *daf-18(ok480); wpEx173[Punc-47::daf-18+myo-2::GFP]*
LX929 *vsIs48[Punc-17::gfp]*
IZ629 *ufIs34[Punc-47::mCherry]*
OAR117 *ufIs34[Punc-47::mCherry; daf-18(ok480)]*
OAR118 *vsIs48[(Punc-17::GFP); daf-18(ok480)]*
OAR142 *ufis34[Punc-47::mCherry; daf-16(mgDf50)]*
OAR143 *ufis34 [Punc-47::mCherry; daf-16(mgDf50); daf-18(ok480)]*
CF1553 *muIs84[(pAD76) Psod-3::gfp + rol-6(su1006)]*
OAR140 *muIs84[(pAD76) Psod-3::gfp + rol-6]; daf-18(ok480)*
OAR141 *muIs84[(pAD76) Psod-3::gfp + rol-6]; daf-16(mgDf50)*
OH99 *mgIs18[Pttx-3::gfp]*
OAR83 *daf-18(ok480); mgIs18[Pttx-3::gfp]*
MT13471 *nIs121[Ptph-1::gfp]*
OAR112 *nIs121[Ptph-1::gfp]; daf-18(ok480)*
IZ805 *ufIs53[Punc-17::ChR2]*
ZM3266 *zxIs3[Punc-47::ChR2::YFP]*
OAR177 *ufIs53[Punc-17::ChR2;;YFP]; daf-18(ok480)*
OAR178 *ufIs53[Punc-17::ChR2::YFP]; daf-16(mgDf50)*
OAR179 *zxIs3[Punc-47::ChR2::YFP];daf-18(ok480)*
OAR180 *zxIs3[Punc-47::ChR2::YFP]; daf-16(mgDf50)*
TJ1052 *age-1(hx546)*
GR1310 *akt-1(mg144)*
GR1318 *pdk-1(mg142)*
JT9609 *pdk-1(sa680)*
VC204 *akt-2(ok393)*
VC222 *raga-1(ok386)*
KQ1366 *rict-1(ft7)*

### Paralysis assays

Paralysis assays were carried out in standard NGM plates with 2 mM aldicarb (Sigma-Aldrich) or 0.5 mM levamisole (Alfa Aesar). 25–30 L4 worms were transferred to each plate and paralyzed animals were counted every 15 or 30 min. An animal was considered paralyzed when it did not respond after prodding three times with a platinum wire on the head and tail (*Blanco et al., 2018*). At least four

independent trials with 25–30 animals for each condition were performed. The area under the curve for each condition in each experiment was used for statistical comparisons.

### Escape response

Escape response assays were performed on NGM agar plates seeded with a thin bacterial lawn of OP50 *E. coli*. To maintain tight control of growth and moisture, 120 µl of bacteria were seeded 24 hr before the assay and grown overnight at 37°C. The day of the assay, young adult worms were transferred to the plates and allowed to acclimate for at least 5 min. Omega turns were induced by gentle anterior touch with fine eyebrow hair and were classified as closed when the worm touched the tail with its head as previously described (*Donnelly et al., 2013*). Between 4 and 7 independent trials with ~20 animals for each condition were performed.

### Body length assays

Body length measurements were performed in standard NGM agar plates without bacteria. Young adult synchronized worms were transferred into the plates and allowed to acclimate for at least 5 min. Worms were recorded with an Amscope Mu300 camera. Animal body length, before and after touching with a platinum pick, was measured using FIJI ImageJ software. Quantification of body shortening after touching was calculated as the decrease of body length related to the length of the animal before being touched.

### Commissure analysis

Synchronized L1 or L4 animals carrying the fluorescence reporters *vsIs48* (*Punc-17::GFP*, cholinergic neurons) or *ufIs34* (*Punc-47::mCherry*, GABAergic neurons) were immobilized with sodium azide (0.25 M) on 2% agarose pads. Commissures of GABAergic and cholinergic neurons were scored with a Nikon Eclipse TE 2000 fluorescence microscope. A commissure is generally composed of a single process, and occasionally two neurites that extend together dorsally. Defects on commissures, including guidance defects, abnormal branching, and incomplete commissures were classified similarly to previous reports (; *Oliver et al., 2019*). The percentage of animals with at least one commissure defect was calculated for each neuronal class (e.g., cholinergic or GABAergic). At least three trials (~20 animals per condition in each trial) were analyzed for each individual experiment. Representative images shown in the figures were collected using laser confocal microscopy (ZEISS LSM 900 with AirScan II) with ×20 and ×63 objectives.

### βHB assays

Worms were exposed to 20 mM DL-3-hydroxybutyric acid sodium salt (Acros Organics) on NGM agar plates seeded with *E. coli* OP50. We synchronized the animals similarly to other experiments by placing gravid animals in NGM plates containing 20 mM of βHB and removing them after 2 hr.

For experiments involving exposure at different developmental stages, animals were transferred between plates with and without βHB as needed. For the earliest exposure, eggs were laid on plates containing 20 mM of βHB and then transferred to drug-free plates to complete their development. Conversely, for later exposures, animals were born on βHB-free plates and subsequently transferred to βHB-containing plates at the specified time (see *Figure 5*).

### *sod-3* expression

*sod-3* expression levels were analyzed in transgenic strains containing the transcriptional reporter *muIs84*, as described previously (*De Rosa et al., 2019*; *Andersen et al., 2022*). Synchronized L4 animals were anesthetized with sodium azide (0.25 M) and mounted on 2% agarose pads. Images were collected using a Nikon Eclipse TE 2000 fluorescence microscope. GFP fluorescence intensity was quantified in same-sized Regions of Interest (ROIs) at the head of the animal using ImageJ FIJI software. Results were normalized to control conditions (wild-type individuals without βHB). ~35–60 animals for each genotype/condition were analyzed.

### Optogenetic assays

We examined young adult animals (6–8 hr post-L4 stage) that express ChR2 in either GABAergic (*Punc-47::ChR2*) or cholinergic neurons (*Punc-17::ChR2*). We transferred these animals to a NGM 6 mm agar

plate without food, let them acclimate for 5 min, and recorded each animal at 15 frames per second using an Allied Vision Alvium 1800 U-500m camera. To stimulate neuronal activity, we exposed the animals to 470 nm light pulses for 5 s. These light pulses were delivered using a custom Python script (VIMBA Peron!; *Garelli, 2023*; available here; *Garelli, 2024*) to an Arduino Uno microcontroller, which operated a Mightex compact universal LED controller (Mightex SLC-MA02-U). The light emission was achieved through a Mightex High-Power LED Collimator Source (LCS-0470-03-11). To precisely track the changes in the worm's body, we continuously monitored its area from 5 s before the light stimulus, during the light stimulus and until 5 s afterward. We developed a FIJI-ImageJ macro capable of automatically tracking the body area in each frame, capitalizing on the clear contrast between the worm's body and the background. As demonstrated in *Videos 3 and 4*, changes in body area directly corresponded to alterations in the animal's length. To compare the changes induced by optogenetic activity between different strains, the body area measurements for each animal were averaged from second 6 (1 s after the blue light was turned on) to second 9 (1 s before the light was turned off). These average values were used for the statistical comparisons detailed in each figure legend.

To validate our measurement system, we manually measured the width of 6 animals at the 2.5 s point of light stimulation and compared them to the body area and length. Our observations consistently showed that, regardless of whether the area increased or decreased (depending on the activation of GABAergic or cholinergic neurons), the width remained mostly unchanged (*Figure 2F, G*). Therefore, the observed changes in the animal's area measured by our FIJI-ImageJ macro indeed represent alterations in the animal's length.

### AIY and HSN analysis

Synchronized L4 or Young Adult worms carrying the fluorescence reporters *Pttx-3::gfp* (AIY interneurons expressing *GFP*) and *Ptph-1::gfp* (HSN expressing *GFP*) were immobilized with sodium azide (0.25 M) on 2% agarose pads and analyzed with a Nikon Eclipse TE 2000 fluorescence microscope. AIY neurons morphology was sorted in qualitative categories (see figure legend) while the migration of HSN was classified in quantitative categories using ImageJ software.

### Statistical analysis

The results presented in each figure are the average of at least three independent assays. Bars represent mean ± standard deviation. Typically, one-way analysis of variance was employed for analyzing multiple parametric samples, and Tukey's *post hoc* test was used for pairwise comparisons among all groups. For comparisons against a control group, Dunnett's *post hoc* test was used. For multiple non-parametric samples, the Kruskal–Wallis test was applied followed by Dunn's *post hoc* test, which was also utilized for comparisons against the control group. In cases where comparisons were made between two independent conditions, a *t*-test was utilized for parametric data, while the Mann–Whitney *U* test was employed for non-parametric data. We used the software GraphPad Prism version 6.01 to perform statistics. The statistical information is indicated in the figure legends. For all assays, the scoring was done blinded. All raw data are available in Open Science Framework (here).

## Acknowledgements

Some strains were provided by the CGC, which is funded by NIH Office of Research Infrastructure Programs (P40 OD010440). We thank Mark Alkema, Michael Francis, and Alex Byrne for strains. We thank Andrés Garelli, Guillermo Spitzmaul, Gabriela Salvador, Mark Alkema, Inés Carrera, Claire Bénard, Alexandra Byrne, and Michael Francis for helpful discussions. In addition, we would like to acknowledge Ignacio Bergé, Adrian Bizet, Carolina Gomila, Marta Stulhdreher, María José Tiecher, Edgardo Buzzi, Susana Gonzalez, and Carla Chrestía for technical support.

# Additional information

## Funding

| Funder | Grant reference number | Author |
|---|---|---|
| Consejo Nacional de Investigaciones Científicas y Técnicas | PUE- N22920170100017CO | Sebastián Giunti<br>María Gabriela Blanco<br>María José De Rosa<br>Diego Rayes |
| Consejo Nacional de Investigaciones Científicas y Técnicas | PIP No. 11220200101606CO | María José De Rosa<br>Diego Rayes |
| Agencia Nacional de Promoción Científica y Tecnológica | PICT 2019-0480 | Diego Rayes |
| Agencia Nacional de Promoción Científica y Tecnológica | PICT-2021-I-A-00052 | Diego Rayes |
| Agencia Nacional de Promoción Científica y Tecnológica | PICT-2017-0566 | María José De Rosa |
| Agencia Nacional de Promoción Científica y Tecnológica | PICT-2020-1734 | María José De Rosa |
| Universidad Nacional del Sur | PGI: 24/B291 | Diego Rayes |
| Universidad Nacional del Sur | PGI: 24/B261 | María José De Rosa |

The funders had no role in study design, data collection, and interpretation, or the decision to submit the work for publication.

## Author contributions

Sebastián Giunti, Data curation, Formal analysis, Validation, Investigation, Visualization, Methodology, Writing – original draft, Writing – review and editing; María Gabriela Blanco, Data curation, Formal analysis, Investigation, Methodology, Writing – original draft; María José De Rosa, Conceptualization, Data curation, Formal analysis, Supervision, Funding acquisition, Investigation, Visualization, Methodology, Writing – original draft, Writing – review and editing; Diego Rayes, Conceptualization, Resources, Data curation, Formal analysis, Supervision, Funding acquisition, Validation, Investigation, Methodology, Writing – original draft, Project administration, Writing – review and editing

## Author ORCIDs

Sebastián Giunti ⓘ https://orcid.org/0009-0009-3608-7041
María Gabriela Blanco ⓘ https://orcid.org/0000-0001-9611-1482
María José De Rosa ⓘ https://orcid.org/0000-0002-2149-6127
Diego Rayes ⓘ https://orcid.org/0000-0002-5448-1431

Reviewer #1 (Public Review): https://doi.org/10.7554/eLife.94520.3.sa1
Reviewer #2 (Public Review): https://doi.org/10.7554/eLife.94520.3.sa2
Reviewer #3 (Public Review): https://doi.org/10.7554/eLife.94520.3.sa3
Author response https://doi.org/10.7554/eLife.94520.3.sa4

# Additional files

## Supplementary files
• MDAR checklist

## Data availability

All raw data are available in Open Science Framework (here).

The following dataset was generated:

| Author(s) | Year | Dataset title | Dataset URL | Database and Identifier |
|---|---|---|---|---|
| De Rosa BG, Rayes | 2024 | The ketone body β-hydroxybutyrate ameliorates neurodevelopmental deficits in the GABAergic system of daf-18/PTEN *Caenorhabditis elegans* mutants | https://osf.io/mdpgc | Open Science Framework, mdpgc |

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
