## [Editor Report · eLife assessment]

This is a conceptually appealing study in which the authors identify genes whose function is **important** for the development of inhibitory (GABA) neurons, and then demonstrate that a diet rich in ketone body β-hydroxybutyrate partially suppresses specific mutant phenotypes. The authors provide **compelling** evidence that features methods, data and analyses more rigorous than the current state-of-the-art. Conceptually, this is evidence of a rescue of a developmental defect with dietary metabolic intervention, linking, in an elegant way, the underpinning genetic mechanisms with novel metabolic pathways that could be used to circumvent the defects.

---

## [Referee Report · Reviewer #1 (Public Review)]

Summary

This interesting study, which has greatly improved in the current revised version, explores the mechanism behind an increased susceptibility of daf-18/PTEN mutant nematodes to paralyzing drugs that exacerbate cholinergic transmission. The authors use state-of-the-art genetics and neurogenetics coupled with locomotor behavior monitoring and neuroanatomical observations using gene expression reporters to show that the susceptibility occurs due to low levels of DAF-18/PTEN in developing inhibitory GABAergic neurons early during larval development (specifically, during the larval L1 stage). DAF-18/PTEN is convincingly shown to act cell-autonomously in these cells upstream of the PI3K-PDK-1-AKT-DAF-16/FOXO pathway, consistent with its well-known role as an antagonist of this conserved signaling pathway. The authors exclude a role for the TOR pathway in this process and present evidence implicating selectivity towards-developing GABAergic neurons of the ventral nerve cord in comparison to excitatory cholinergic neurons. Finally, the authors show that a diet supplemented with a ketogenic body, β-hydroxybutyrate, which also counteracts the PI3K-PDK-1-AKT pathway, promoting DAF-16/FOXO activity, partially rescues the proper development (morphology and function) of GABAergic neurons in daf-18/PTEN mutants, but only if the diet is provided early during larval development. This strongly suggests that the critical function of DAF-18/PTEN in developing inhibitory GABAergic neurons is to prevent excessive PI3K-PDK-1-AKT activity during this critical and particularly sensitive period of their development in juvenile L1 stage worms. Whether or not the sensitivity of GABAergic neurons to DAF-18/PTEN function is a defining and widespread characteristic of this class of neurons in *C. elegans* and other animals, or rather a particularity of the early developmental stage of the GABAergic neurons investigated remains to be determined.

Strengths:

The study reports interesting and important findings, advancing the knowledge of how daf-18/PTEN and the PI3K-PDK-1-AKT pathway can influence neurodevelopment, and providing a valuable paradigm to study the selectivity of gene activities towards certain neurons. It also defines a solid paradigm to study the potential of dietary interventions (such as ketogenic diets) or other drug treatments to counteract (prevent or revert?) neurodevelopment defects and stimulate DAF-16/FOXO activity.

The fact that other non-GABAergic *C. elegans* neurons (i.e., AIY and HSN neurons) are also sensitive to DAF-18/PTEN activity during development suggests that the particular sensitivity observed in the GABAergic ventral nerve cord neurons in this study could be unrelated to their neurotransmitter class (GABAergic) per se, but rather to some other neuronal property (a critical period of plasticity or activity-based wiring?) that these neurons share with the AIY and HSN neurons, and not with the other surveyed ventral nerve cord neurons (the excitatory cholinergic neurons). The relevance of this possibility within the framework of understanding the role of DAF-18/PTEN in E/I imbalance across clades is not fully clear at this stage.

---

## [Referee Report · Reviewer #2 (Public Review)]

Summary:

Disruption of the excitatory/inhibitory (E/I) balance have been reported in Autism Spectrum disorders (ASD) to which PTEN mutations have been associated. Giunti et al choose to explore the impact of PTEN mutations on the balance between E/I signaling using as a platform the *C. elegans* neuromuscular system where both cholinergic (E) and GABAergic (I) motor neurons regulate muscle contraction and relaxation. Mutations in daf-18/PTEN specifically affect morphologically and functionally the GABAergic (I) system, while leaving the cholinergic (E) system unaffected. The study further reveals that the observed defects in the GABAergic system in daf-18/PTEN mutants are attributed to reduced activity of DAF-16/FOXO during development.

Moreover, ketogenic diets (KGDs), known for their effectiveness in disorders associated with E/I imbalances such as epilepsy and ASD, are found to induce DAF-16/FOXO during early development. Supplementation with β-hydroxybutyrate in the nematode at early developmental stages proves to be both necessary and sufficient to correct the effects on GABAergic signaling in daf-18/PTEN mutants.

Strengths:

The authors combined pharmacological, behavior and optogenetic experiments to show the GABAergic signaling impairment at the *C. elegans* neuromuscular junction in DAF-18/PTEN and DAF-16/FOXO mutants. Moreover, by studying the neuron morphology, they point towards neurodevelopmental defects in the GABAergic motoneurons involved in locomotion. Using the same set of experiments, they demonstrate that a ketogenic diet can rescue the inhibitory defect in the daf-18/PTEN mutant at an early stage.

Weaknesses:

The morphological experiments hint towards a pre-synaptic defect to explain the GABAergic signaling impairment, but it would have also been interesting to check the post-synaptic part of the inhibitory neuromuscular junctions such as the GABA receptor clusters to assess if the impairment is only presynaptic or both post and presynaptic. Moreover, analysing post-synaptic functionality in-depth using electrophysiology would be beneficial too.

Nevertheless, this question alone could be entirely the subject of another paper and is not essential to the primary message of the paper.

Conclusion:

Giunti et al provide fundamental insights into the connection between PTEN mutations and neurodevelopmental defects through DAF-16/FOXO and shed light on the mechanisms through which ketogenic diets positively impact neuronal disorders characterized by E/I imbalances.

---

## [Referee Report · Reviewer #3 (Public Review)]

Summary:

This is a conceptually appealing study by Giunti et al in which the authors identify a role for PTEN/daf-18 and daf-16/FOXO in the development of inhibitory GABA neurons, and then demonstrate that a diet rich in ketone body β-hydroxybutyrate partially suppresses the PTEN mutant phenotypes. The authors use three assays to assess their phenotypes: (1) pharmacological assays (with levamisole and aldicarb); (2) locomotory assays and (3) cell morphological assays. These assays are carefully performed and the article is clearly written. While neurodevelopmental phenotypes had been previously demonstrated for PTEN/daf-18 and daf-16/FOXO (in other neurons), and while KB β-hydroxybutyrate had been previously shown to increase daf-16/FOXO activity (in the context of aging), this study is significant because it demonstrates the importance of KB β-hydroxybutyrate and DAF-16 in the context of neurodevelopment. Conceptually, and to my knowledge, this is the first evidence I have seen of a rescue of a developmental defect with a dietary metabolic intervention, linking, in an elegant way, the underpinning genetic mechanisms with novel metabolic pathways that could be used to circumvent the defects.

---

## [Author Response]

The following is the authors’ response to the original reviews.

**Public Reviews:**

**Reviewer #1 (Public Review):**
Summary:This interesting study explores the mechanism behind an increased susceptibility of daf-18/PTEN mutant nematodes to paralyzing drugs that exacerbate cholinergic transmission. The authors use state-of-theart genetics and neurogenetics coupled with locomotor behavior monitoring and neuroanatomical observations using gene expression reporters to show that the susceptibility occurs due to low levels of DAF-18/PTEN in developing inhibitory GABAergic neurons early during larval development (specifically, during the larval L1 stage). DAF-18/PTEN is convincingly shown to act cell-autonomously in these cells upstream of the PI3K-PDK-1-AKT-DAF-16/FOXO pathway, consistent with its well-known role as an antagonist of this conserved signaling pathway. The authors exclude a role for the TOR pathway in this process and present evidence implicating selectivity towards developing GABAergic neurons. Finally, the authors show that a diet supplemented with a ketogenic body, β-hydroxybutyrate, which also counteracts the PI3K-PDK-1-AKT pathway, promoting DAF-16/FOXO activity, partially rescues the proper development (morphology and function) of GABAergic neurons in daf-18/PTEN mutants, but only if the diet is provided early during larval development. This strongly suggests that the critical function of DAF18/PTEN in developing inhibitory GABAergic neurons is to prevent excessive PI3K-PDK-1-AKT activity during this critical and particularly sensitive period of their development in juvenile L1 stage worms. Whether or not the sensitivity of GABAergic neurons to DAF-18/PTEN function is a defining and widespread characteristic of this class of neurons in *C. elegans* and other animals, or rather a particularity of the unique early-stage GABAergic neurons investigated remains to be determined.Strengths:The study reports interesting and important findings, advancing the knowledge of how daf-18/PTEN and the PI3K-PDK-1-AKT pathway can influence neurodevelopment, and providing a valuable paradigm to study the selectivity of gene activities towards certain neurons. It also defines a solid paradigm to study the potential of dietary interventions (such as ketogenic diets) or other drug treatments to counteract (prevent or revert?) neurodevelopment defects and stimulate DAF-16/FOXO activity.Weaknesses:(1) Insufficiently detailed methods and some inconsistencies between Figure 4 and the text undermine the full understanding of the work and its implications.The incomplete methods presented, the imprecise display of Figure 4, and the inconsistency between this figure and the text, make it presently unclear what are the precise timings of observations and treatments around the L1 stage. What exactly do E-L1 and L1-L2 mean in the figure? The timing information is critical for the understanding of the implications of the findings because important changes take place with the whole inhibitory GABAergic neuronal system during the L1 stage into the L2 stage. The precise timing of the events such as neuronal births and remodelling events are welldescribed (e.g., Figure 2 in Hallam and Jin, Nature 1998; Fig 7 in Mulcahy et al., Curr Biol, 2022). Likewise, for proper interpretation of the implication of the findings, it is important to describe the nature of the defects observed in L1 larvae reported in Figure 1E - at present, a representative figure is shown of a branched commissure. What other types of defects, if any, are observed in early L1 larvae? The nature of the defects will be informative. Are they similar or not to the defects observed in older larvae?

We thank the reviewer for highlighting these areas for improvement. We have updated and clarified the timing of observation in the text, figures, and methodology section accordingly.

All experiments were conducted using age-synchronized animals. Gravid worms were placed on NGM plates and removed after two hours. The assays were then carried out on animals that hatched from the eggs laid during this specific timeframe.

Regarding the detailed timings outlined in the original Figure 4 (now Figure 5 in the revised version), we provided the following information in the revised version: For experiments involving continuous exposure to βHB throughout development, the gravid worms were placed on NGM plates containing the ketone body and removed after two hours. Therefore, this exposure covered the ex-utero embryonic development period up to the L4-Young adult stage when the experiments were conducted.

In experiments involving exposure at different developmental stages as those depicted in Figure 4 of the original version, (now Figure 5, revised version), animals were transferred between plates with and without βHB as required. We exposed daf-18/PTEN mutant animals to βHB-supplemented diets for 18-hour periods at different developmental stages (Figure 5A, revised version). The earliest exposure occurred during the 18 hours following egg laying, covering ex-utero embryonic development and the first 8-9 hours of the L1 stage. The second exposure period encompassed the latter part of the L1 stage, the entire L2 stage, and most of the L3 stage. The third exposure spanned the latter part of the L3 stage (~1-2 hours), the entire L4 stage, and the first 6-7 hours of the adult stage.

All this information has been conveniently included in Figure 5, text (Page13, lines 259-276), and in methodology (Page 4, Lines 85-90, Revised Methods and Supplementary information) of the revised manuscript.

In response to the reviewer's suggestion, we have also included photos of daf-18 worms at the L1 stage (30 min/1h post-hatching). Defects are already present at this early stage, such as handedness and abnormal branching commissures, which are also observed in adult worm neurons (see Supplementary Figure 4, revised version).

These defects manifest in DD neurons shortly after larval birth. The prevalence of animals with errors is higher in L4 worms (when both VDs and DDs are formed) compared to early L1s (Figures 3 C-E and Supplementary Figure 4, revised version). This suggests that defects in VD neurons also occur in daf-18 mutants. Indeed, when we analyzed the neuronal morphology of several wild-type and daf-18 mutant animals, we found defects in the commissures corresponding to both DD and VD neurons (Supplementary Figure 3, revised version).

These data are now included in the revised version (Results (Page 10, lines 177-196), Discussion (Pages 14-16), Main Figure 3, and Supplementary Figures 3, 4 and 7 revised version)

(2) The claim of proof of concept for a reversal of neurodevelopment defects is not fully substantiated by data.The authors state that the work "constitutes a proof of concept of the ability to revert a neurodevelopmental defect with a dietary intervention" (Abstract, Line 56), however, the authors do not present sufficient evidence to distinguish between a "reversal" or prevention of the neurodevelopment defect by the dietary intervention. This clarification is critical for therapeutic purposes and claims of proof-of-concept. From the best of my understanding, reversal formally means the defect was present at the time of therapy, which is then reverted to a "normal" state with the therapy. On the other hand, prevention would imply an intervention that does not allow the defect to develop to begin with, i.e., the altered or defective state never arises. In the context of this study, the authors do not convincingly show reversal. This would require showing "embryonic" GABAergic neuron defects or showing convincing data in newly hatched L1 (0-1h), which is unclear if they do so or not, as I have failed to find this information in the manuscript. Again, the method description needs to be improved and the implications can be very different if the data presented in Figure 2D-E regard newly born L1 animals (0-1h) or L1 animals at say 5-7h after hatching. This is critical because the development of the embryonically-born GABAergic DD neurons, for instance, is not finalized embryonically. Their neurites still undergo outgrowth (albeit limited) upon L1 birth (see DataS2 in Mulcahy et al., Curr Biol 2022), hence they are susceptible to both committing developmental errors and to responding to nutritional interventions to prevent them. In contrast to embryonic GABAergic neurons, embryonic cholinergic neurons (DA/DB) do not undergo neurite outgrowth post-embryonically (Mulcahy et al., Curr Biol 2022), a fact which could provide some mechanistic insight considering the data presented. However, neurites from other post-embryonically-born neurons also undergo outgrowth postembryonically, but mostly during the second half of the L1 stage following their birth up to mid-L2, with significant growth occurring during the L1-L2 transition. These are the cholinergic (VA/VB and AS neurons) and GABAergic (VD) neurons. The fact that AS neurons undergo a similar amount of outgrowth as VD neurons is informative if VD neurons are or are not susceptible to daf-18/PTEN activity. Independently, DD neurons are still quite unique on other aspects (see below), which could also bring insight into their selective response.Finally, even adjusting the claim to "constitutes a proof-of-concept of the ability of preventing a neurodevelpmental defect with a dietary intervention" would not be completely precise, because it is unclear how much this work "constitutes a proof of concept". This is because, unless I misunderstood something, dietary interventions are already applied to prevent neurodevelopment defects, such as when folic acid supplementation is recommended to pregnant women to prevent neural tube defects in newborns.

Thank you very much for pointing out this issue and highlighting the need to further investigate the ameliorative capacity of βHB on GABAergic defects in daf-18 mutants. In the revised version, we have included experiments to address this point.

Our microscopy analyses strongly indicate that the development of DD neurons is affected, with errors observed as early as one-hour post-hatching (Main Figure 3, and Supplementary Figures 4 and 7, revised version). Additionally, based on the position of the commissures in L4s, our results strongly suggest that VD neurons are also affected (Supplementary Figure 3, revised version). Both, the frequency of animals with errors and the number of errors per animal are higher in L4s compared to L1 larvae (Main Figures 3, and Supplementary Figure 4 and 7, revised version). It is very likely that the errors in VD neurons, which are born in the late L1 stage, are responsible for the higher frequency of defects observed in L4 animals.

As the reviewer noted, GABAergic DD neurons, which are born embryonically, do not complete their development during the embryonic stages. Some defects in DD neurons may arise during the postembryonic period. Following the reviewer's suggestion, we analyzed L1 larvae at different times before the appearance of VDs (1 hour post-hatching and 6 hours post-hatching). We did not observe an increase in error prevalence, suggesting that DD defects in daf-18 mutants are mostly embryonic (Supplementary Fig 4B, Revised Version).

Our findings suggest that βHB's enhancement is not due to preventive effects in DDs, as defects persist in newly hatched larvae regardless of βHB presence (Supplementary Figure 7, revised version), and postembryonic DD growth does not introduce new errors (Supplementary Figure 4, revised version). This lack of preventive effect could be due to βHB's limited penetration into the embryonic environment. Unlike early L1s, significant improvement occurs in L4s upon βHB early exposure (Supplementary Figure 7, revised version). This could be explained by a reversing effect on malformed DD neurons and/or a protective influence on VD neuron development. While we cannot rule out the first option, even if all errors in DDs in L1 were repaired (which is very unlikely), it wouldn't explain the level of improvement in L4 (Supplementary Figure 7, revised version). Therefore, we speculate that VDs may be targeted by βHB. The notion that exposure to βHB during early L1 can ameliorate defects in neurons primarily emerging in late L1s (VDs) is intriguing. We may hypothesize that residual βHB or a metabolite from prior exposure could forestall these defects in VD neurons. Notably, βHB has demonstrated a capacity for long-lasting effects through epigenetic modifications (Reviewed in He et al, 2023, https://doi.org/10.1016%2Fj.heliyon.2023.e21098). More work is needed to elucidate the underlying fundamental mechanisms regarding the ameliorating effects of βHB supplementation. We have now discussed these possibilities under discussion (Page 17, lines 369-383, revised version).

We agree with the reviewer that the term "reversal" is not accurate, and we have avoided using this terminology throughout the text. Furthermore, in the title, we have decided to change the word "rescue" to "ameliorate," as our experiments support the latter term but not the former. Additionally, the reviewer is correct that folic acid administration to pregnant women is already a metabolic intervention to prevent neural tube defects. In light of this, we have avoided claiming this as proof of concept in the revised manuscript

(3) The data presented do not warrant the dismissal of DD remodeling as a contributing factor to the daf-18/PTEN defects.Inhibitory GABAergic DD neurons are quite unique cells. They are well-known for their very particular property of remodeling their synaptic polarity (DD neurons switch the nature of their pre- and postsynaptic targets without changing their wiring). This process is called DD remodeling. It starts in the second half of the L1 stage and finishes during the L2 stage. Unfortunately, the fact that the authors find a specific defect in early GABAergic neurons (which are very likely these unique DD neurons) is not explored in sufficient detail and depth. The facts that these neurons are not fully developed at L1, that they still undergo limited neurite growth, and that they are poised for striking synaptic plasticity in a few hours set them apart from the other explored neurons, such as early cholinergic neurons, which show a more stable dynamics and connectivity at L1 (see Mulcahy et al., Curr Biol 2022).The authors use their observation that daf-18/PTEN mutants present morphological defects in GABAergic neurons prior to DD remodeling to dismiss the possibility that the DAF-18/PTEN-dependent effects are "not a consequence of deficient rearrangement during the early larval stages". However, DD remodeling is just another cell-fate-determined process and as such, its timing, for instance, can be affected by mutations in genes that affect cell fates and developmental decisions, such as daf-18 and daf-16, which affect developmental fates such as those related with the dauer fate. Specifically, the authors do not exclude the possibility that the defects observed in the absence of either gene could be explained by precocious DD remodeling. Precocious DD remodeling can occur when certain pathways, such as the lin-14 heterochronic pathway, are affected. Interestingly, lin-14 has been linked with daf16/FOXO in at least two ways: during lifespan determination (Boehm and Slack, Science 2005) and in theL1/L2 stages via the direct negative regulation of an insulin-like peptide gene ins-33 (Hristova et al., Mol Cell Bio 2005). It is likely that the prevention of DD dysfunction requires keeping insulin signaling in check (downregulated) in DD neurons in early larval stages, which seems to coincide with the critical timing and function of daf-18/PTEN. Hence, it will be interesting to test the involvement of these genes in the daf-18/daf-16 effects observed by the authors.

This is another interesting point raised by the reviewer. We have demonstrated that defects manifest in early L1 (30 min-1 hour post-hatching) which corresponds to a pre-remodeling time in wild-type worms.

We acknowledge the possibility of early remodeling in specific mutants as pointed out by the reviewer.

However, the following points suggest that the effects of these mutations may extend beyond the particularity of DD remodeling: (i) Our experiments also show defects in VD neurons in daf-18 mutants (Supplementary Figure 3, revised version), as discussed in our previous response. These neurons do not undergo significant remodeling during their development. (ii) DAF-18 and DAF-16 deficiencies produce neurodevelopmental alteration on other Non-Remodeling Neurons: Severe neurite defects in neurons that are nearly fully formed at larval hatching, such as AIY in daf-18 and daf-16 mutants, have been previously reported (Christensen et al., 2011). Additionally, the migration of another neuron, HSN, is severely affected in these mutants (Kennedy et al., 2013). (iii) To the best of our knowledge, DD remodeling only alters synaptic polarity without forming new commissures or significant altering the trajectory of the formed ones. Thus, it is unlikely (though not impossible) for remodeling defects to cause the observed commissural branching and handedness abnormalities in DD neurons. Therefore, we think that the impact of daf-18 mutations on GABAergic neurons is not primarily linked to DD remodeling but extends to various neuron types. It is intriguing and requires further exploration in the future, the apparent resilience of cholinergic motor neurons to these mutations. This resilience is not limited to daf18/PTEN animals since mutants in certain genes expressed in both neuron types (such as neuronal integrin ina-1 or eel-1, the *C. elegans* ortholog of HUWE1) alter the function or morphology of GABAergic neurons but not cholinergic motor neurons (Kowalski, J. R. et al. Mol Cell Neurosci 2014; Oliver, D. et al. J Dev Biol (2019); Opperman, K. J. et al. Cell Rep 2017). These points are discussed in the manuscript (Discussion, page 15, lines 311-322, revised version) and reveal the existence of compensatory or redundant mechanisms in these excitatory neurons, rendering them much more resistant to both morphological and functional abnormalities.

Discussion on the impact of the work on the field and beyond:The authors significantly advance the field by bringing insight into how DAF-18/PTEN affects neurodevelopment, but fall short of understanding the mechanism of selectivity towards GABAergic neurons, and most importantly, of properly contextualizing their findings within the state-of-the-art *C. elegans* biology.For instance, the authors do not pinpoint which type of GABAergic neuron is affected, despite the fact that there are two very well-described populations of ventral nerve cord inhibitory GABAergic neurons with clear temporal and cell fate differences: the embryonically-born DD neurons and the postembryonically-born VD neurons. The time point of the critical period apparently defined by the authors (pending clarifications of methods, presentation of all data, and confirmation of inconsistencies between the text and figures in the submitted manuscript) could suggest that DAF-18/PTEN is required in either or both populations, which would have important and different implications. An effect on DD neurons seems more likely because an image is presented (Figure 2D) of a defect in an L1 daf-18/PTEN mutant larva with 6 neurons (which means the larva was processed at a time when VD neurons were not yet born or expressing pUnc-47, so supposedly it is an image of a larva in the first half of the L1 stage 0-~7h?). DD neurons are also likely the critical cells here because the neurodevelopment errors are partially suppressed when the ketogenic diet is provided at an "early" L1 stage, but not later (e.g., from L2-L3, according to the text, L2-L4 according to the figure?).

Thank you for this insightful input. As previously mentioned, we conducted experiments in this revision to clarify the specificity of GABAergic errors in daf-18/PTEN mutants, in particular, whether they affect DDs, VDs, or both. Our results suggest that commissural defects are not limited to DD neurons but also occur in VD neurons (Supplementary Figure 3). Regarding the effect of βHB, our findings suggest that VD neurons are targets of βHB action. As mentioned in the previous response and the discussion section (Page 17, lines 369-383, revised version), we might speculate that lingering βHB or a metabolite from prior exposure could mitigate these defects in VD neurons that are born in Late L1s-Early L2s. Additionally, βHB has been noted for its capacity to induce long-term epigenetic changes. Therefore, it could act on precursor cells of VD neurons, with the resulting changes manifesting during VD development independently of whether exposure has ceased. All these possibilities are now discussed in the manuscript.

Acknowledging that our work raises several questions that we aim to address in the future, we believe our manuscript provides valuable information regarding how the PI3K pathway modulates neuronal development and how dietary interventions can influence this process.

This study brings important contributions to the understanding of GABAergic neuron development in *C. elegans*, but unfortunately, it is justified and contextualized mostly in distantly-related fields - where the study has a dubious impact at this stage rather than in the central field of the work (post-embryonic development of *C. elegans* inhibitory circuits) where the study has stronger impact. This study is fundamentally about a cell fate determination event that occurs in a nutritionally-sensitivedevelopmental stage (post-embryonic L1 larval stage) yet the introduction and discussion are focused on more distantly related problems such as excitatory/inhibitory (E/I) balance, pathophysiology of human diseases, and treatments for them. Whereas speculation is warranted in the discussion, the reduced indepth consideration of the known biology of these neurons and organisms weakens the impact of the study as redacted. For instance, the critical role of DAF-18/PTEN seems to occur at the early L1 larval stage, a stage that is particularly sensitive to nutritional conditions. The developmental progression of L1 larvae is well-known to be sensitive to nutrition - eg, L1 larvae arrest development in the absence of food, something that is explored in nematode labs to synchronize animals at the L1 stage by allowing embryos to hatch into starvation conditions (water). Development resumes when they are exposed to food. Hence, the extensive postembryonic developmental trajectory that GABAergic neurons need to complete is expected to be highly susceptible to nutrition. Is it? The sensitivity towards the ketogenic diet intervention seems to favor this. In this sense, the attribution of the findings to issues with the nutrition-sensitive insulin-like signaling pathway seems quite plausible, yet this possibility seems insufficiently considered and discussed.

We greatly appreciate the reviewer's emphasis on the sensitivity of the L1 stage to nutritional status. As the reviewer points out, *C. elegans* adjusts its development based on food availability, potentially arresting development in L1 in the absence of food. It is therefore reasonable that both the completion of DD neuron trajectories and the initial development steps of VD neurons are particularly sensitive to dietary modulation of the insulin pathway, in which both DAF-18 and DAF-16 play roles. This important point has also been included in the discussion (Page 18, lines 384-407, revised version).

Finally, the fact that imbalances in excitatory/inhibitory (E/I) inputs are linked to Autism Spectrum Disorders (ASD) is used to justify the relevance of the study and its findings. Maybe at this stage, the speculation would be more appropriate if restricted to the discussion. In order to be relevant to ASD, for instance, the selectivity of PTEN towards inhibitory neurons should occur in humans too. However, at present, the E/I balance alteration caused by the absence of daf-18/PTEN in *C. elegans* could simply be a coincidence due to the uniqueness of the post-embryonic developmental program of GABAergic neurons in *C. elegans*. To be relevant, human GABAergic neurons should also pass through a unique developmental stage that is critically susceptible to the PI3K-PDK1-AKT pathway in order for DAF18/PTEN to have any role in determining their function. Is this the case? Hence, even in the discussion, where the authors state that "this study provides universally relevant information on.... the mechanisms underlying the positive effects of ketogenic diets on neuronal disorders characterized by GABA dysfunction and altered E/I ratios", this claim seems unsubstantiated as written particularly without acknowledging/mentioning the criteria that would have to be fulfilled and demonstrated for this claim to be true. Our results suggest that defects in GABAergic neurons are not limited to DDs, which, as the reviewer rightly notes, are quite unique in their post-embryonic development primarily due to the synaptic remodeling process they undergo. These defects also extend to VD neurons, which do not exhibit significant developmental peculiarities once they are born. Therefore, we think that the defects are not specific to the developmental program of DD neurons but are more related to all GABAergic motoneurons. Additionally, the observation of defects in non-GABAergic neurons in *C. elegans* daf-18 mutants supports the hypothesis that the role of daf-18 is not limited to DD neurons (Christensen et al., 2011; Kennedy et al., 2013). In mammals, Pten conditional knockout (cKO) animals have been extensively studied for synaptic connectivity and plasticity, revealing an imbalance between synaptic excitation and inhibition (E/I balance) (Reviewed in Rademacher and Eickholt, 2019, Cold Spring Harbor Perspect Med, https://doi.org/10.1101%2Fcshperspect.a036780). This imbalance is now widely accepted as a key pathological mechanism linked to the development of ASD-related behavior (Lee et al, 2017; Biological Psychiatry, https://doi.org/10.1016/j.biopsych.2016.05.011). The importance of PTEN in the development of GABAergic neurons in mammals is well-documented. For instance, embryonic PTEN deletion from inhibitory neurons impacts the establishment of appropriate numbers of parvalbumin and somatostatin-expressing interneurons, indicating a central role for PTEN in inhibitory cell development (Vogt et al, 2015, Cell Rep, https://doi.org/10.1016%2Fj.celrep.2015.04.019). Additionally, conditional PTEN knockout in GABAergic neurons is sufficient to generate mice with seizures and autism-related behavioral phenotypes (Shin et al, 2021, Molecular Brain, https://doi.org/10.1186%2Fs13041-02100731-8). Moreover, while mice in which PV GABAergic neurons lacked both copies of Pten experienced seizures and died, heterozygous animals (PV-Pten+/−) showed impaired formation of perisomatic inhibition (Baohan et al, 2016, Nature Comm, OI: 10.1038/ncomms12829). Therefore, there is substantial evidence in mammals linking PTEN mutations to neurodevelopmental disorders in general and affecting GABAergic neurons in particular. Hence, we believe that the role of daf-18/PTEN in GABAergic development could be a more widespread phenomenon across the animal kingdom rather than a specific process unique to *C. elegans*.

Beyond the points discussed, we have addressed the reviewer's comment regarding the last sentence of the abstract. We have revised it to more cautiously frame the relationship between our findings, ASD, and mammalian neurodevelopmental disorders.

**Reviewer #2 (Public Review):**
Summary:Disruption of the excitatory/inhibitory (E/I) balance has been reported in Autism Spectrum Disorders(ASD), with which PTEN mutations have been associated. Giunti et al choose to explore the impact of PTEN mutations on the balance between E/I signaling using as a platform the *C. elegans* neuromuscular system where both cholinergic (E) and GABAergic (I) motor neurons regulate muscle contraction and relaxation. Mutations in daf-18/PTEN specifically affect morphologically and functionally the GABAergic (I) system, while leaving the cholinergic (E) system unaffected. The study further reveals that the observed defects in the GABAergic system in daf-18/PTEN mutants are attributed to reduced activity of DAF-16/FOXO during development.Moreover, ketogenic diets (KGDs), known for their effectiveness in disorders associated with E/I imbalances such as epilepsy and ASD, are found to induce DAF-16/FOXO during early development. Supplementation with β-hydroxybutyrate in the nematode at early developmental stages proves to be both necessary and sufficient to correct the effects on GABAergic signaling in daf-18/PTEN mutants.Strengths:The authors combined pharmacological, behavioral, and optogenetic experiments to show theGABAergic signaling impairment at the *C. elegans* neuromuscular junction in DAF-18/PTEN and DAF-16/FOXO mutants. Moreover, by studying the neuron morphology, they point towardsneurodevelopmental defects in the GABAergic motoneurons involved in locomotion. Using the same set of experiments, they demonstrate that a ketogenic diet can rescue the inhibitory defect in the daf18/PTEN mutant at an early stage.Weaknesses:The morphological experiments hint towards a pre-synaptic defect to explain the GABAergic signaling impairment, but it would have also been interesting to check the post-synaptic part of the inhibitory neuromuscular junctions such as the GABA receptor clusters to assess if the impairment is only presynaptic or both post and presynaptic.Moreover, all observations done at the L4 stage and /or adult stage don't discriminate between the different GABAergic neurons of the ventral nerve cord, ie the DDs which are born embryonically and undergo remodeling at the late L1 stage, and VDs which are born post-embryonically at the end of the L1 stage. Those additional elements would provide information on the mechanism of action of the FOXO pathway and the ketone bodies.

Thank you for your insightful suggestions.

This is an initial study that serves as a cornerstone, demonstrating the sensitivity of GABAergic neuron development to alterations in the PI3K pathway and how these alterations can be mitigated by a dietary intervention with a ketone body. While we have determined that the transcription factor DAF-16/FOXO is essential in the neurodevelopmental process and is the target of ketone bodies to alleviate defects, there are still underlying mechanisms to be elucidated. This is only the first step that opens many avenues for further investigation, including the study of post-synaptic partners.

While our current study primarily focuses on neuronal alterations without delving into potential postsynaptic effects, we do plan to investigate this aspect in future research. This includes examining GABAergic receptors as well as cholinergic receptors, as exacerbation of cholinergic signaling cannot be ruled out. To conduct a comprehensive study of post-synaptic structure and functionality, we would need strains with fluorescent markers for both pre- and post-synaptic components (such as rab-3, unc-49, unc29, acr-16 fusion to GFP or mCherry). Unfortunately, most of these strains are not currently available in our laboratory. Unlike the US or Europe, acquiring these strains from the *C. elegans* CGC repository in Argentina is challenging due to common customs delays, which require significant time and resources to navigate. Discussions at the Latin American *C. elegans* conference with CGC administrators, such as Ann Rougvie, have been initiated to address this issue, but a solution has not been reached yet. Additionally, to analyze post-synaptic functionality in-depth, studying the response to perfusion with various agonists using electrophysiology would be beneficial. We are in the process of acquiring the capability to conduct electrophysiology experiments in our laboratory, but progress is slow due to limited funding.

While we believe these experiments are very informative, they will require a considerable amount of time due to our current circumstances. We consider them non-essential to the primary message of the paper, which focuses on neuronal developmental defects leading to functional alterations in daf-18/PTEN mutants and the novel finding that these can be mitigated by supplementing food with hydroxybutyrate. We will study the structure and functionality of the post-synapse in our future projects and also plan to extend this investigation to mutants with deficiencies in genes closely related to neurodevelopmental defects, such as neuroligin, neurexin, or shank-3, which have been implicated in synaptic architecture.

We also agree that discriminating between DD and VD neurons provides significant insights into the neurodevelopmental phenomena dependent on the FOXO pathway and the action of βHB. In this revised version, we present evidence that not only DD neurons are affected but also VD neurons (see Supplementary Figure 3, revised version). This allows us to suggest that daf-18 affects the development of GABAergic neurons regardless of whether they are born embryonically (DDs) or post-embryonically (VDs) (see also our response to the previous reviewer). We hope to distinguish the defects observed in each type of neuron in future studies. For this, we would need to use strains specifically marked in one neuronal type or another, which, for the same reasons mentioned earlier, would take a considerable amount of time under current conditions.

Conclusion:Giunti et al provide fundamental insights into the connection between PTEN mutations and neurodevelopmental defects through DAF-16/FOXO and shed light on the mechanisms through which ketogenic diets positively impact neuronal disorders characterized by E/I imbalances.
**Reviewer #3 (Public Review):**
Summary:This is a conceptually appealing study by Giunti et al in which the authors identify a role for PTEN/daf-18 and daf-16/FOXO in the development of inhibitory GABA neurons, and then demonstrate that a diet rich in ketone body β-hydroxybutyrate partially suppresses the PTEN mutant phenotypes. The authors use three assays to assess their phenotypes: (1) pharmacological assays (with levamisole and aldicarb; 2) locomotory assays and (3) cell morphological assays. These assays are carefully performed and the article is clearly written. While neurodevelopmental phenotypes had been previously demonstrated for PTEN/daf-18 and daf-16/FOXO (in other neurons), and while KB β-hydroxybutyrate had been previously shown to increase daf-16/FOXO activity (in the context of aging), this study is significant because it demonstrates the importance of KB β-hydroxybutyrate and DAF-16 in the context of neurodevelopment. Conceptually, and to my knowledge, this is the first evidence I have seen of a rescue of a developmental defect with dietary metabolic intervention, linking, in an elegant way, the underpinning genetic mechanisms with novel metabolic pathways that could be used to circumvent the defects.Strengths:What their data clearly demonstrate, is conceptually appealing, and in my opinion, the biggest contribution of the study is the ability of reverting a neurodevelopmental defect with a dietary intervention that acts upstream or in parallel to DAF-16/FOXO.Weaknesses:The model shows AKT-1 as an inhibitor of DAF-16, yet their studies show no differences from wildtype in akt-1 and akt-2 mutants. AKT is not a major protein studied in this paper, and it can be removed from the model to avoid confusion, or the result can be discussed in the context of the model to clarify interpretation.

Thank you very much for the suggestion. We agree with the reviewer's appreciation that the study of AKT's action itself is too limited in this study to draw conclusions that would allow its inclusion in the proposed model. Therefore, following the reviewer's suggestion, we have removed this protein from our model

When testing additional genes in the DAF-18/FOXO pathway, there were no significant differences from wild-type in most cases. This should be discussed. Could there be an alternate pathway via DAF-18/DAF16, excluding the PI3K pathway or are there variations in activity of PI3K genes during a ketogenic diet that are hard to detect with current assays?

Thank you for bringing up this point. Our pharmacological experiments indeed demonstrate that all mutants associated with an exacerbation of the PI3K pathway, which typically inhibits nuclear translocation and activity of the transcription factor DAF-16, lead to imbalances in E/I

(excitation/inhibition) that manifest as hypersensitivity to cholinergic drugs. This includes the gain of function of pdk-1 and the loss of function of daf-18 and daf-16 itself. In our subsequent experiments, we demonstrate that this exacerbation of the PI3K pathway leads to errors in the neurodevelopment of GABAergic neurons, which explains the hypersensitivity to aldicarb and levamisole.

As the reviewer remarks, it is intriguing why mutants inhibiting this pathway do not show differences in their sensitivity to cholinergic drugs compared to wild-type animals. We can speculate, for instance, that during neurodevelopment, there is a critical period where the PI3K pathway must remain with very low activity (or even deactivated) for proper development of GABAergic neurons. This could explain why there are no differences in sensitivity to cholinergic drugs between mutants that inhibit the PI3K pathway and the wild type. The PI3K pathway depends on insulin-like signals, which are in turn positively modulated by molecules associated with the presence of food. Interestingly, larval stage 1 is particularly sensitive to nutritional status, being able to completely arrest development in the absence of food. Therefore, dietary intervention with BHB may generate a signal of dietary restriction (as seen in mammals) and, as a consequence of this dietary restriction, the PI3K pathway is inhibited, resulting in increased DAF-16 activity. This could restore the proper neurodevelopment of GABAergic neurons. However, this is mere speculation, and further deeper experiments (than the pharmacology ones we performed here) with mutants in different genes within the PI3K pathway may shed light on this point.

Following the reviewer's suggestion, this point has been discussed in the revised version of the manuscript. (Discussion Page 18, Lines 384-407).

The consequence of SOD-3 expression in the broader context of GABA neurons was not discussed. SOD3 was also measured in the pharynx but measuring it in neurons would bolster the claims.

SOD-3 is a known target of DAF-16. Previous studies have shown that βHB induces SOD-3 expression through the induction of DAF-16 (Edwards et al, 2014, Aging, https://doi.org/10.18632%2Faging.100683). The highest levels of SOD-3 expression are typically observed in the pharynx or intestine (DeRosa et al, 2019 https://doi.org/10.1038/s41586-019-1524-5; Zheng et al., 2021, PNAS, https://doi.org/10.1073/pnas.2021063118), and it is often used as a measure of general upregulation of DAF-16. Therefore, we used this parameter as a measure of βHB upregulating systemic DAF-16 activity. While we agree with the reviewer that observing variations in SOD-3 expression in neurons would further support our conclusions, unfortunately, we did not detect measurable signals of SOD-3 in motor neurons in either the control condition or the daf-18 background even upon stress or BHB-exposure. This may be because SOD-3 is a minor target of DAF-16 in these neurons, or its modulation may not correspond to the timing of fluorescence measurements (L4-adults).

Despite this, our genetic experiments and neuron-specific rescue experiments lead us to conclude that DAF-16 must act autonomously in GABAergic neurons to ensure proper neurodevelopment.

If they want to include AKT-1, seeing its effect on SOD-3 expression could be meaningful to the model.

Thank you for this suggestion. We believe that even measuring SOD-3 levels in akt mutant backgrounds would still provide limited information to give it a predominant value in our work. Additionally, to have a complete understanding of the total role of AKT, it would be necessary to measure it in a double mutant background of akt-1; akt-2, and these double mutants generate 100 % dauers even at 15C (Oh et al., PNAS 2005, https://doi.org/10.1073/pnas.0500749102; Quevedo et al., Current Biology 2007, http://dx.doi.org/10.1016/j.cub.2006.12.038; Gatzi et al., PLOS ONE 2014, https://doi.org/10.1371/journal.pone.0107671), greatly complicating the execution of these experiments. Therefore, following the first advice of this reviewer, we have decided to modify our model by excluding AKT.

**Recommendations for the authors:**

**Reviewer #1 (Recommendations For The Authors):**
⁃ Please include earlier in the main text the rationale for using unc-25 as a control/reference already when mentioning Figure 1A.

Thank you for pointing out the need to reference this control earlier. We have included the following paragraph in the description of Figure 1 (Page 5, line 71, revised version):

“Hypersensitivity to cholinergic drugs is typical of animals with an increased E/I ratio in the neuromuscular system, such as mutants in unc-25 (the *C. elegans* orthologue for glutamic acid decarboxylase, an essential enzyme for synthesizing GABA). While daf-18/PTEN mutants become paralyzed earlier than wild-type animals, their hypersensitivity to cholinergic drugs is not as severe as that observed in animals completely deficient in GABA synthesis, such unc-25 null mutants (Figures 1B and 1C) indicating a less pronounced imbalance between excitatory and inhibitory signals.”

⁃ Please discuss the greater sensitivity of pdk-1(gf) animals to levamisole than to aldicarb.

Thank you for bringing up this subtle point. We understand that the reviewer is referring to the paralysis curve in response to aldicarb in pdk-1(gf), which is closer to unc-25 than the curve for levamisole (in both cases, they are more sensitive than the wild type). Therefore, pdk-1(gf) animals seem to be more sensitive to aldicarb than to levamisole. These results are now shown in Figure 1D (revised version).

The PI3K pathway does not only act in neurons but also in muscles. Gain of function in pdk-1 has been shown to modulate muscle protein degradation (Szewczyk et al, EMBO Journal, 2008. https://doi.org/10.1038/sj.emboj.7601540). In contrast, no effect on protein degradation has been reported for null mutants in this gene. Several studies have demonstrated that protein degradation levels can differentially affect receptor subunits, particularly acetylcholine receptors (Reviewed in Crespi et al, Br J Pharmacol, 2018). *C. elegans* is characterized by a wide repertoire of AChR subunits, and there are at least two subtypes of ACh receptors in muscles (one multimeric sensitive to levamisole and one homomeric (ACR-16) insensitive to levamisole) (Richmond et al, 1999 Nature Neuroscience http://dx.doi.org/10.1038/12160; Touroutine D, JBC 2005 https://doi.org/10.1074/jbc.M502818200).

Interestingly, acr-16 null mutants are hypersensitive to aldicarb (Zeng et al, JCB, 2023, https://doi.org/10.1083/jcb.202301117) while the electrophysiological response to levamisole in this mutant remains similar to that of wild-type (Tourorutine et al, 2005). Therefore, it may be that the gain of function in pdk-1 induces a change in the expression of AChR subtypes in muscle that differentially affect sensitivity to levamisole and ACh. This is purely speculative, and there may be many other explanations. While it would be interesting to explore this difference further, it goes far beyond the scope of this study. The cholinergic drug sensitivity assay is purely exploratory and allowed us to delve into the GABAergic and cholinergic signals in daf-18 mutants. In this sense, the hypersensitivity of pdk-1(gf) to both drugs supports the idea that an increase in PI3K signaling leads to an increased E/I ratio.

Please explain the rationale to perform akt-1 and akt-2 assays separated. Why not test doublemutants? Has their lack of redundancy been determined?.Our pharmacological assays are conducted at the L4 larval stage, making it impossible to analyze the potential redundancy of akt-1 and akt-2 in sensitivity to levamisole and aldicarb. This impossibility arises because the akt-1;akt-2 double mutant exhibits nearly 100% arrest as dauer even at 15°C, as reported in several prior studies (Oh et al., PNAS 2005, https://doi.org/10.1073/pnas.0500749102; Quevedo et al., Current Biology 2007, http://dx.doi.org/10.1016/j.cub.2006.12.038; Gatzi et al., PLOS ONE 2014, https://doi.org/10.1371/journal.pone.0107671). While the increased dauer arrest in the double mutant compared to the single mutants might suggest redundant functions in dauer entry, there are also reports indicating the absence of redundancy in other processes, such as vulval development (Nakdimon et al., PLOS Genetics 2012, https://doi.org/10.1371%2Fjournal.pgen.1002881).

The complete Dauer arrest likely underlies why other studies focusing on the role of the PI3K pathway in neurodevelopment utilize both mutants separately (Christensen et al, Development 2011, https://doi.org/10.1242/dev.069062). While determining the potential redundancy of these genes is not feasible for this assay, we utilized various mutants of the pathway (age-1, pdk-1, daf-18, daf-16 and daf16;daf-18 in addition to the akt-s) that support the conclusion, which is that exacerbating the PI3K pathway activity makes animals hypersensitive to cholinergic drugs.

In response to the reviewer's concern, we have added a sentence in the text explaining the impossibility of performing the assay in the akt-1;akt-2 double mutant (Page 6, lines90-92)

Figure 1C and D (This applies to all similarly presented bar figures). Please show data points and dispersion (preferably data, median+- 25-75% or average+-SD).

Thank you. Done

⁃ Line 112 -maybe "and resumes"?

Thank you. Done (Line 126, revised version)

⁃ Figure 1E and F. Please present mean +-SD (not SEM) of fluctuations. Please change slightly the tones so that the dispersion is easier to distinguish on the "blue light on" box.

For the revised version, we have also included bar graphs for each optogenetic experiment, representing the mean of the length average of each worm measured from the first second after the blue light was turned on until the second before the light was turned off (in the graph, this corresponds to the period between seconds 6 and 9 of the traces). These graphs include the standard deviation and the corresponding significance levels. All of this has been included in the new legend (Figure 2D, 2E, 4E-J).Figure 1A&1B & Supplementary Figure 1D x Supplementary Figure 1E&1F. What is the difference between these experiments? Whereas the unc-25 mutants paralyze in the same amount of time, the WT animals paralyze ~1 h later in Supplementary Figure 1E-1F in response to either drug. Please revise experimental conditions to see if anything can be learned eg, maybe this is a nutritional response from experiments done at different timepoints? Maybe different food recipes affected sensitivity to paralysis?

Thank you for pointing this out. While the experiments with daf-18 (in both alleles) and daf-16 were conducted at the beginning of this project (2019-2020), the assays with the other mutants in the PI3K and mTOR pathways were performed years later. Changes in the reagents used (agar, peptone, cholesterol, etc.) to grow the worms have occurred, potentially altering the animals' response directly or through the nutritional quality of the bacteria they grow on. In addition, the difference may be attributed to the fact that experiments at the project's outset were conducted by one author, while more recent experiments were carried out by another. The goal is to quantify paralysis in non-responsive worms after touch stimulation. The force of this probing or the thickness of the hair used for touching can be slightly operator-dependent and can lead to variable responses. In addition, always the presence of wild-type and unc-25 strain is included as internal control in every experiment. Nevertheless, despite this userdependent variation, the experiments were always conducted blindly (except for unc-25, whose uncoordinated phenotype is easily identifiable), thus we trust in the outcomes.

Supplementary Figure 1G - Length and Width appear to be switched in both left and right panels - please revise and include a description of N and of statistics depicted.

Unfortunately, we don't see the switching error that the reviewer mentioned. In the left panel, we demonstrate that optogenetic activation of GABAergic neurons leads to an increase in length without modifying the width of the animal. Therefore, we conclude that the increase in area, as observed in our Fiji macro for optogenetic response analysis, is due to an increase in the animal's length. In the cholinergic activation shown in the right panel, the animal shortens (decreasing length) without modifying the width, resulting in the reduction of the total body area.

We have included information about N (sample size) and the statistical test used in the legends as suggested. These graphs are now shown as Figures 2F and G, revised version.

Supplementary Figure 1G legend lines 779-780. Please describe the post-hoc test applied following ANOVA to obtain the denoted p values. This applies to all datasets where ANOVA or Krusal-Wallis tests were applied.

Following reviewer´s suggestion, all the post-hoc tests applied after ANOVA or Kruskal-Wallis analysis were included in the legend of each figure and Materials and Methods (statistical analysis section).

Line 174 maybe "arises *from* the hyperactivation" instead of *for*?.

Corrected. Thank you. Line 190, revised version.

Supplementary Figure 4. On line 816 it says n=40-90, but please check the n of the daf-18, daf-16 samples, which seem to have less than 40 animals.

We understand that the reviewer is referring to Supplementary Figure 3 from the original version (now Supplementary Figure 5 in the revised version). We have now included the number of observations below each data point cloud to clearly indicate the sample size for each condition

⁃ Supplementary Figure 4 - please state what are the bars on the graphs. Please state which post-hoc test was performed after Kruskal-Wallis and present at least the p values obtained between treated controls and each genotype. Alternatively, present the whole truth table in supplementary daita.

We understand that the reviewer is referring to Supplementary Figure 3 from the original version (now Supplementary Figure 5 in the revised version). There was an error in the original legend (thank you for bringing this to our attention) since the statistics were not performed using Kruskall-Wallis in this case, but rather each treated condition was compared to its own untreated control using Mann-Whitney test. We have now added the p-values to the graph. All raw data for this figure, as well as for all other figures, are available in Open Science Framework (https://osf.io/mdpgc/?view_only=3edb6edf2298421e94982268d9802050).

Please cite the figure panels in order: eg, Figure 3E is mentioned in the text after panels Figure 3F-K.

Done. We have rearranged the figures to adapt them to the text order (Figure 4, revised version)

Figure 4 - line 610 please revise "(n=20-30 n: 20-25 animals per genotype/trial)."

Thank you. Corrected.

Figure 4 - there appears to be an inconsistency in the figure with the text (lines 223-225). In figures it says E-L1, but in the text, it says "solely in L1". Does E-L1 include the whole L1 stage? If not- E-L1 can be interpreted only as during the embryonic stage, hence, no exposure to betaHB due to the impermeable chitin eggshell. Then there is L1-L2, which should cover the L1 stage and the L2 or something else. Please revise. The text mentions L2-L3 or L3-L4 and these categories are not in the figures. This clarification is key for the interpretation of the results. The precise developmental time of the exposures is not defined either in the methods or in the figures. Please provide precise times relative to hours and/or molts and revise the text/figure for consistency.

The reviewer is entirely correct in pointing out the lack of relevant data regarding the exposure time to βHB. We have now clarified the information For the revised version, we have adjusted the nomenclature of each exposure period to precisely reflect the developmental stages involved.

For the experiments involving continuous exposure to βHB throughout development, the NGM plate contained the ketone body. Therefore, the exposure encompassed, in principle, the ex-utero embryonic development period up to L4-Young adults (E-L4/YA, in Figure 5A) when the experiments were conducted. Since it could be a restriction to drug penetration through the chitin shell of the eggs (see Supplementary Figure 7), we can ensure βHB exposure from hatching.

In experiments involving exposure at different developmental stages as those depicted in Figure 4 of the original version, (now Figure 5), animals were transferred between plates with and without βHB as required. We exposed daf-18/PTEN mutant animals to βHB-supplemented diets for 18-hour periods at different developmental stages (Figure 5A). The earliest exposure occurred during the 18 hours following egg laying, covering ex-utero embryonic development and the first 8-9 hours of the L1 stage (This period is called E-L1, in figure 5 revised version). The second exposure period encompassed the latter part of the L1 stage, the entire L2 stage, and most of the L3 stage (L1-L3). The third exposure spanned the latter part of the L3 stage (~1-2 hours), the entire L4 stage, and the first 6-7 hours of the adult stage (L3-YA).

All this information has been conveniently included in Figure 5 (and its legend), text (Page 13, lines 259276), and Material and Methods of the revised manuscript.

Some methods are not sufficiently well described. Specifically, how the animals were exposed to treatments and how stages were obtained for each experiment. Was synchronization involved? If so, in which experiments and how exactly was it performed?

As mentioned in previous responses all the experiments were performed in age-synchronized animals. We include the following sentence in Materials and Methods (*C. elegans* culture and maintenance section): “All experiments were conducted on age-synchronized animals. This was achieved by placing gravid worms on NGM plates and removing them after two hours. The assays were performed on the animals hatched from the eggs laid in these two hours”.

**Reviewer #2 (Recommendations For The Authors):**
Major points(1) To complete the study on the GABAergic signaling at the NMJs, it would be interesting to assess the status of the post-synaptic part of the synapse such as the GABAR clustering. It would also tell if the impairment is only presynaptic or both post and presynaptic.

Thank you for your insightful suggestion. We agree that exploring post-synaptic elements can shed light on whether the impairment is solely presynaptic or involves both pre and post-synaptic components.

While our current study primarily focuses on neuronal alterations without delving into potential postsynaptic effects, we do plan to investigate this aspect in the future. This includes not only examining GABAergic receptors but also exploring cholinergic receptors, as exacerbation of cholinergic signaling cannot be ruled out. To conduct a comprehensive study of post-synaptic structure and functionality, we would need strains with fluorescent markers for both pre and post-synaptic components (rab-3, unc-49, unc-29, acr-16 driving GFP or mCherry). However, most of these strains are not currently available in our laboratory. Unlike the US or Europe, acquiring these strains from the *C. elegans* CGC repository in Argentina is challenging due to common customs delays, requiring significant time and resources to navigate. Discussions at the Latin American *C. elegans* conference with CGC administrators, such as Ann Rougvie, have been initiated to address this issue, but a solution has not been reached yet.

Additionally, to analyze post-synaptic functionality in-depth, studying the response to perfusion with various agonists using electrophysiology would be beneficial. We are in the process of acquiring the capability to conduct electrophysiology experiments in our laboratory, but progress is slow due to limited funding.

While we believe these experiments are very informative, they will require a considerable amount of time due to our current circumstances. We consider them non-essential to the primary message of the paper, which focuses on neuronal morphological defects leading to functional alterations in daf-18/PTEN mutants.

We will include these experiments in our future projects, also planning to extend this investigation to mutants with deficiencies in genes closely related to neurodevelopmental defects, such as neuroligin, neurexin, or shank-3, which have been implicated in synaptic architecture.

(2) The author always referred to unc-47 promoter or unc-17 promoter, never specifying where those promoters are driving the expression (and in the Materials & Methods, no information on the corresponding sequence). Depending on the promoters they may not only be expressed in the motoneurons involved in locomotion (VA, VB, DA, DB, VD, and DD), but they could also be expressed in other neurons which could be of importance for the conclusions of the optogenetic assays but also the daf-18 expression in GABAergic neurons. We appreciate the reviewer's insight regarding the broader expression patterns of the unc-17 and unc-47 promoters in all cholinergic and GABAergic neurons, respectively. The strains expressing constructs with these promoters were obtained from the CGC or other labs and have been widely used in previous papers (Liewald et al, Nature Methods, https://www.nature.com/articles/nmeth.1252 2008); Byrne, A. B. et al. Neuron 81, 561-573, doi:10.1016/j.neuron.2013.11.019 (2014).

Regarding the optogenetic assays, the readout utilized (body length elongation or contraction) is primarily associated with the activity of cholinergic and GABAergic motor neurons and has been used in numerous studies to measure motor neuron functionality (Liewald et al, Nature Methods, https://www.nature.com/articles/nmeth.1252 (2008);Hwang, H. et al. Sci Rep 6, 19900, doi:10.1038/srep19900 (2016); Schultheis et al, . J Neurophysiol 106, 817-827, doi:10.1152/jn.00578.2010 (2011); Koopman, M., Janssen, L. & Nollen, E. A. BMC Biol 19, 170, doi:10.1186/s12915-021-01085-2 (2021);). It has previously been established that the shortening observed after optogenetic activation of the unc-17 promoter, while active in various interneurons, depends on the activity of cholinergic motor neurons (Liewald et al., Nature Methods, https://www.nature.com/articles/nmeth.1252 (2008)). This was demonstrated by examining transgenic worms expressing ChR2-YFP from another cholinergic, motoneuronspecific but weaker promoter, Punc-4. They observed contraction and coiling upon illumination, albeit to a milder degree.

In terms of GABAergic neurons, only 3 do not directly synapse to body wall muscles (AVL, PDV, and RIS) and are primarily involved in defecation. Of the 23 GABAergic motor neurons, 19 are Dtype motoneurons, while the remaining 4 innervate head muscles (Pereira et al, eLife 2015, https://doi.org/10.7554/eLife.12432). It is therefore expected that while there may be some contribution from these latter neurons to the elongation after optogenetic activation in animals containing punc-47::ChR2, the main contribution should be from the D-type neurons. Additionally, while there may be some influence on D-type neuron development due to daf-18 rescue in neurons like RME, DVB or AVL, the most direct explanation for the rescue is that daf-18 acts autonomously in D-type cells. Additionally, we have pharmacological and behavioral assays that support the findings of optogenetics and enable us to reach final conclusions.

(3) DD neurons are born during embryogenesis and newborn L1s have neurites even though less than at a later stage. If possible, it would be interesting to take a look at them to see if βHB has an effect or not. It will corroborate the hypothesis that βHB action is prevented by the impermeable eggshell on a system that can respond at a later stage. Moreover, using a specific DD, DA, and DB promoter, it would be possible to check if there is a difference in the morphological defects between embryonic and post-embryonic neurons.

This is a very interesting point raised by the reviewer. We conducted experiments to analyze the morphology of GABAergic neurons in animals exposed to βHB only during the ex-utero embryonic development (in their laid egg state). We observed that this incubation was not sufficient to rescue the defects in GABAergic neurons (Supplementary Figure 7, revised version). As reported by other authors and discussed in our paper, the chitinous eggshell might act as an impermeable barrier to most drugs. However, we cannot rule out that incubation during this period is necessary but not sufficient to mitigate the defects. We have included these experiments in Supplementary Figure 7 and in the text (Page 13, lines 272-276)

Additionally, we analyzed confocal images where, based on their position, we could identify and assess errors in DD (embryonic) and VD (Post-embryonic) neurons (Supplementary Figure 3, revised version). These experiments show that the effects are observed in both types of neurons, and we did not observe any differential alterations in neuronal morphology between the two types of neurons.

Minor points(1) Expression of daf-18/PTEN in muscle or hypodermis, could it ensure a proper development? It could give insights into the action mechanism of βHB.

The reviewer's observation is indeed very intriguing. Previous studies from the Grishok lab (Kennedy et al, 2013) have demonstrated that the expression of daf-18 or daf-16 in extraneuronal tissues, specifically in the hypodermis, can rescue migratory defects in the serotoninergic neuron HSN in daf-18 or daf-16 null mutants of *C. elegans*. Clearly, this could also be an option for rescuing the morphological and functional defects of GABAergic motoneurons.

However, the fact that the expression of daf-18 in GABAergic neurons rescues these defects strongly suggests an autonomous effect. In this regard, autonomous effects of DAF-18 or DAF-16 on neurodevelopmental defects have also been reported in interneurons in *C. elegans* (Christensen et al, 2011). This is included in the discussion (Page 15, lines 330-335)

(2) Re-organise the introduction. The paragraph on ketogenic diets (lines 35-38) is not logically linked.

Following reviewer´s suggestion we have reorganized the introduction and changed the order of explanation regarding the significance of ketogenic diets, linking it with their proven effectiveness in alleviating symptoms of diseases with E/I imbalance (Lines 23-60, revised version)

(3) Incorporate titles in the result section to guide the reader.

Done. Thank you

(4) Systematically add PTEN or FOXO when daf-18 or daf-16 are mentioned (for example lines 69, 84, 85).

Done. Thank you

(5) Strain lists: lines 646 to 653: some information is missing on the different transgenes used in this study (integrated (Is) or extrachromosomal (Ex) with their numbers).

Thank you for bringing this to our attention. We have now included all the information regarding the different transgenes used in this study, including whether they are integrated (Is) or extrachromosomal (Ex) and their respective numbers. This information can be found in the revised version of the manuscript (Materials and Methods, *C. elegans* culture and maintenance section highlighted in yellow).

**Reviewer #3 (Recommendations For The Authors):**
In Figure 1, some experiments were done with the unc-25 control while others, such as the optogenetic experiments, were done without those controls.

Thank you for pointing this out. In the optogenetic experiments, we waited for the worm to move forward for 5 seconds at a sustained speed before exposing it to blue light to standardize the experiment, as the response can vary if the animal is in reverse, going forward, or stationary. Due to the severity of the uncoordinated movement in unc-25 mutants, achieving this forward movement before exposure is very difficult. Additionally, this lack of coordination prevents these animals from performing the escape response tests, as they barely move. Therefore, we limited the use of this severe GABAergic-deficient control to pharmacological or post-prodding shortening experiments.